# Sensory cortex wiring requires preselection of short- and long-range projection neurons through an Egr-Foxg1-COUP-TFI network

Pei-Shan Hou[1,2,3], Goichi Miyoshi [4] & Carina Hanashima[1,2,3]

The bimodal requisite for a genetic program and external stimuli is a key feature of sensory circuit formation. However, the contribution of cell-intrinsic codes to directing sensory-specific circuits remains unknown. Here, we identify the earliest molecular program that preselects projection neuron types in the sensory neocortex. Mechanistically, Foxg1 binds to an H3K4me1-enriched enhancer site to repress *COUP-TFI*, where ectopic acquisition of Foxg1 in layer 4 cells transforms local projection neurons to callosal projection neurons with pyramidal morphologies. Removal of Foxg1 in long-range projection neurons, in turn, derepresses COUP-TFI and activates a layer 4 neuron-specific program. The earliest segregation of projection subtypes is achieved through repression of Foxg1 in layer 4 precursors by early growth response genes, the major targets of the transforming growth factor-β signaling pathway. These findings describe the earliest cortex-intrinsic program that restricts neuronal connectivity in sensory circuits, a fundamental step towards the acquisition of mammalian perceptual behavior.

[1] Department of Biology, Faculty of Education and Integrated Arts and Sciences, Waseda University, Shinjuku-ku, Tokyo 162-8480, Japan. [2] Graduate School of Advanced Science and Engineering, Waseda University, Shinjuku-ku, Tokyo 162-8480, Japan. [3] Laboratory for Neocortical Development, RIKEN Center for Developmental Biology, Kobe 650-0047, Japan. [4] Department of Neurophysiology, Tokyo Women's Medical University School of Medicine, Shinjuku-ku, Tokyo 162-8666, Japan. Correspondence and requests for materials should be addressed to C.H. (email: hanashima@waseda.jp)

Since the discovery of Brodmann areas[1], protomap[2] and protocortex[3] hypotheses have argued the contribution of intrinsic vs. extrinsic regulation in establishing sensory-specific circuits in the cerebral cortex. In these models, the protomap model supports the cortex-intrinsic origin of area specification, whereas the protocortex model argues that afferent projections relayed through the thalamus lie central to the establishment of sensory areas. At the cellular level, sensory inputs received by layer 4 (L4) granular neurons are delivered locally to layer 2/3 (L2/3) intracortical projection neurons, which finally propagate to subcortical targets through long-range axons of layer 5 (L5) subcerebral projection neurons (SCPNs)[4,5]. Thus, L4 neurons serve as a central gateway for establishing sensory circuits to initiate higher order information processing.

While these two models propose complementary roles in sensory circuit formation[2,6,7], recent work has emphasized the importance of extrinsic regulation, in which manipulation of inputs through thalamic innervation[8,9], neuronal activity[10], and cell positioning[11] perturb the number of L4 neurons generated. Theoretically, however, recruitment of thalamic axons to their precise targets must involve a preselection mechanism of cortical layer neurons to guide these axon terminals, implying that a cortex-intrinsic mechanism underlies L4 development. Consistent with this view, blockage of sensory innervation[6], neuronal transmitter depletion[12], and rerouting of thalamic axons[13] all fail to eliminate Rorβ-expressing neurons, a hallmark of L4 thalamic input neurons. Furthermore, alteration in cell distribution results in correct L4 neuron specification and sensory circuit activation[9], implying that the identity of L4 neurons is established independent of their extracellular environment. However, to date, the intrinsic machinery that preselects L4 neurons from long-range projection neurons to restrict their competence to attract thalamic innervation has remained unknown.

In contrast to the poor understanding of the ontogeny of L4 neurons, studies over the last decade have uncovered a core gene regulatory network that delineates long-range projection neuron subtypes of the neocortex[14]. In particular, corticofugal projection neurons (CFuPNs) and callosal projection neurons (CPNs) are produced through sequential derepression of key transcription factors (TFs) expressed in the developing neocortex[14–19]. In this scheme, the onset of Foxg1 expression in progenitor cells suppresses a default transcriptional network to initiate a SCPN-specific program[20,21]. Subsequent transition from SCPN to CPN production is achieved by activation of Satb2 expression, which is mediated through SCPN-derived feedback signals[17,20,21]. Consequently, these gene networks have revealed transcriptional cascades that direct the sequential production of layer 1, 6, 5, and 2/3 neurons in the neocortex; however, L4 sensory input neurons have been excluded from this network. Indeed, in contrast to long-range CFuPNs[22] and CPNs[17], TFs unique to thalamic input L4 local projection neurons are underrepresented, with the exception of the orphan nuclear receptor Rorβ, the only common marker identified for L4 neurons. Rorβ manipulation affects Brn1/2 expression[23] and thalamic innervation[24]; however, the onset of Rorβ expression in L4 neurons appears relatively late during corticogenesis[25]. Moreover, inhibition of thalamocortical innervation does not eliminate Rorβ expression in L4 neurons[6,8], indicating that the induction of Rorβ requires a cortex-intrinsic regulatory mechanism.

In this study, we found that the coordination between two highly evolutionarily conserved TFs susceptible to neurological disorders, the forkhead-box family gene Foxg1 and the nuclear hormone receptor COUP-TFI/Nr2f1, reciprocally regulates the earliest specification of long- and short-range projection neurons in the neocortex. Ectopic expression of Foxg1 or loss of COUP-TFI expression attenuates the acquisition of L4 fate and induces a

long-range projection neuron-specific program. COUP-TFI gain-of-function (GOF), in turn, alters distal projection neurons to acquire a L4 neuron identity. We further demonstrate that downregulation of Foxg1 is the key event for L4 neuron specification mediated through COUP-TFI. This early segregation of projection subtypes is achieved through repression of Foxg1 by early growth response (Egr) genes, the major target of the transforming growth factor-β (TGFβ) signaling pathway activated in L4 precursors. These results identify the earliest intrinsic program that confers unique laminar and hodological properties in the neocortex, a fundamental step for the formation of sensory-specific circuits.

## Results

### Reciprocal COUP-TFI and Foxg1 expression in laminar subtypes.
To reveal the gene regulatory network that delineates thalamic input local projection neurons from long-range projection neurons, we performed an in silico screen for TFs expressed in a complementary manner between these laminar subtypes. The transcriptomic profile across cortical layers in the mouse cortex[26] (http://genserv.anat.ox.ac.uk/layers) revealed enriched Foxg1 expression in L2/3 CPNs and L5 long-range SCPNs but low expression in thalamic input (L4) neurons (Fig. 1a), indicating its differential roles in the development of long (layers 2/3/5) vs. short-range (L4) projection neurons. To assess the repression targets of Foxg1 that distinguish between these projection types, we utilized transcriptome data that manipulated Foxg1 expression in vivo during corticogenesis[20] (Fig. 1b–d). Among the significantly downregulated genes upon Foxg1 induction (Fig. 1c, d), COUP-TFI/Nr2f1, an evolutionarily conserved orphan nuclear receptor gene, was found among the cluster of genes that exhibited the fastest and significant downregulation in gene expression (Fig. 1d). We next examined the expression of Foxg1 and COUP-TFI at the mid-corticogenesis period, which demonstrated mutual expression at E15.5 (Fig. 1e–e″). Temporal dynamics of Foxg1 and COUP-TFI expression showed that at E11.5, Foxg1 was mainly detected in progenitor cells of the ventricular zone (VZ), whereas COUP-TFI was expressed in a subpopulation of preplate cells (Fig. 1f–f″)[27]. Notably, at the cellular level, cells with high COUP-TFI expression exhibited low or no Foxg1 expression (Fig. 1f–f″). At E13.5, COUP-TFI was scattered in the VZ and weakly expressed in some progenitor cells, whereas Foxg1 was broadly expressed in the progenitor cells. In the cortical plate (CP), cells in the most superficial region of the CP expressed high levels of COUP-TFI, whereas other cells that expressed Foxg1 showed low or no COUP-TFI expression (Fig. 1g–g″). Immunohistochemistry detected Foxg1-negative COUP-TFI-positive cells in the marginal zone as in earlier stages, whereas Foxg1 and COUP-TFI were coexpressed in the deeper portion of the CP (Fig. 1h–h″). In contrast, double immunohistochemistry/in situ hybridization revealed that Foxg1 mRNA is absent in subplate and layer 6 corticothalamic projection neurons (CThPNs) (Fig. 1e–e″), indicating the perdurance of Foxg1 protein but lack of transcription activation in this population. Notably, many COUP-TFI-positive Foxg1-negative cells were detected in the intermediate zone at this stage (Fig. 1e–e″, g–h″). On postnatal day (P)1, when neurogenesis has completed but L2/3 cortical neurons are still migrating, Foxg1 was widely expressed in CP neurons at variable levels but absent in Cajal–Retzius cells in the marginal zone and SP neurons (Fig. 1i–i″). At P4, when all projection neurons have arrived in the CP, L2/3 cells expressed high Foxg1 and low COUP-TFI. Notably, Foxg1 and COUP-TFI showed complementary expression in L5 neurons, in which the lower part of L5 cells (L5b) expressed high Foxg1 with low or no COUP-TFI expression, and the upper part of L5 cells

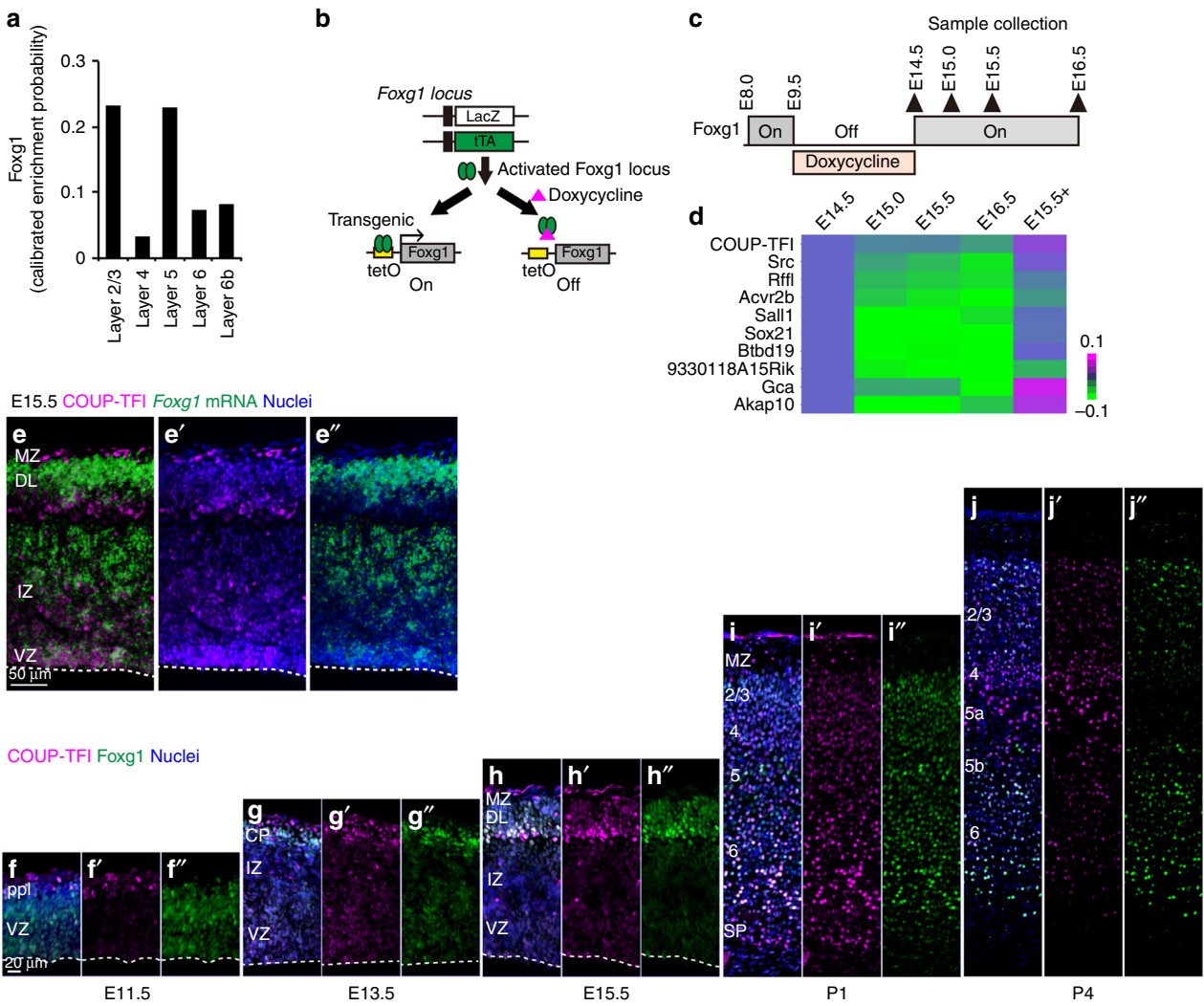

**Fig. 1** Reciprocal expression of COUP-TFI and Foxg1 in the developing neocortex. **a** Calibrated enrichment probability for Foxg1 expression across cortical layers in the adult mouse somatosensory cortex (http://genserv.anat.ox.ac.uk). **b** Schematic diagram showing the strategy of Foxg1 expression manipulation by doxycycline administration. In the absence of doxycycline, tet-transactivator (tTA) protein binds to tetO promoter to activate Foxg1 transgene expression. In the presence of doxycycline, doxycycline binds to tTA protein to prevent the activation of transgenic Foxg1 expression. **c** Schematic diagram showing the timing of doxycycline administration and the corresponding Foxg1 expression. Samples were collected at indicated time points shown in closed arrowheads. **d** Hierarchical clustering using the complete linkage method with Euclidean distance. Heatmap represents the gene cluster that exhibited rapid expression downregulation upon Foxg1 induction by doxycycline administration. **e–e″** Complementary expression of Foxg1 mRNA (green) by in situ hybridization and COUP-TFI protein (red) immunohistochemistry in E15.5 mouse cortex. Dashed lines indicate the ventricular surface. **f–j″** Developmental expression of COUP-TFI (red) and Foxg1 (green) in E11.5 (**f–f″**), E13.5 (**g–g″**), E15.5 (**h–h″**), P1 (**i–i″**), and P4 (**j–j″**) wild-type cortices. Mouse anti-COUP-TFI (Perseus) and Rabbit anti-Foxg1 (TaKaRa) antibodies were used. Embryonic tissues were processed at the identical condition using cryosections and postnatal tissues were perfused prior to fixation using floating sections. Dashed lines indicate the ventricular surface. VZ ventricular zone, IZ intermediate zone, CP cortical plate, MZ marginal zone, DL deep layer

(L5a) expressed low or no Foxg1 but high COUP-TFI expression. L4 cells were significantly enriched in COUP-TFI expression, whereas only the upper-most L4 cells expressed Foxg1 (Fig. 1j–j″). Thus, Foxg1 and COUP-TFI exhibit dynamic and complementary expression in cortical precursors and postmitotic neurons, indicating their reciprocal function in cortical laminar subtypes.

We next examined whether this mutual expression of Foxg1 and COUP-TFI is achieved through cell-autonomous or non-cell autonomous mechanisms. We introduced Foxg1 into the developing neocortex and analyzed these neurons 1 day after in utero electroporation (IUE) (Fig. 2a). Immunostaining revealed significant decrease in COUP-TFI expression in Foxg1-overexpressed cells (Fig. 2b, c). In contrast, conditional removal of Foxg1 using Emx1$^{Cre/+}$ mice[28] showed increased COUP-TFI

expression in EGFP-positive cells (Foxg1 cKO in Fig. 2d, e), suggesting that COUP-TFI is negatively regulated by Foxg1 in neocortical precursor cells.

We next examined the regulatory machinery responsible for this rapid response of COUP-TFI to Foxg1 expression and their mutual expression in cortical precursors. Based on previous Foxg1-ChIP sequencing analysis using embryonic mouse neocortex (Fig. 2f)[20], we identified two putative Foxg1 binding sites (PBS1 and PBS2) within and near the COUP-TFI gene (Fig. 2f). We then performed Foxg1-qChIP analysis using E14.5 cortices and demonstrated specific binding of Foxg1 to one of the putative binding sites, PBS1, using Foxg1 knockout neocortical cells for a comparison (Fig. 2g). Notably, this region was enriched with H3K4me1, indicating an enhancer-like property (Fig. 2g). To

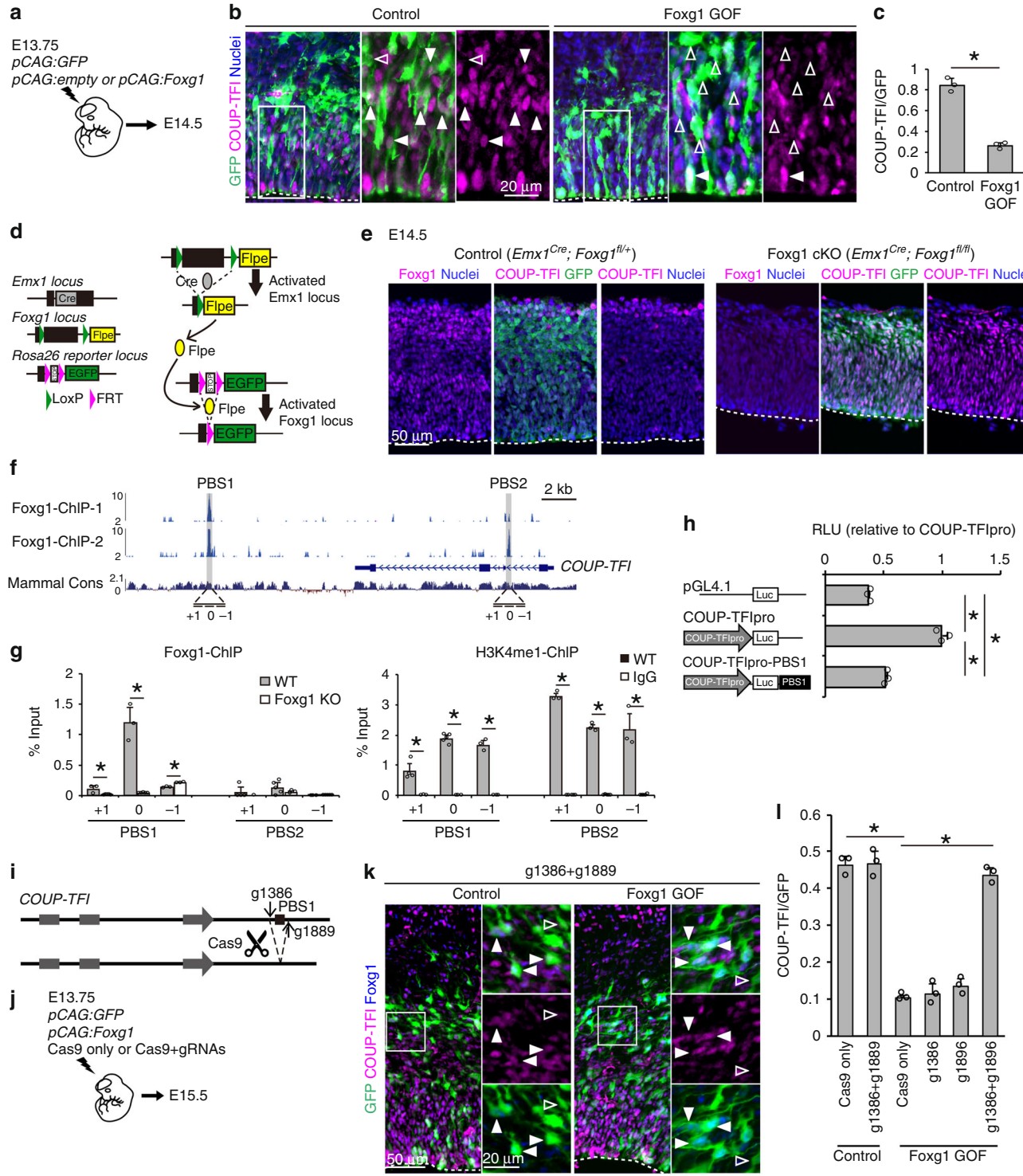

assess whether this enhancer-like region, PBS1, is targeted by Foxg1 to regulate the activity of the *COUP-TFI* promoter, we performed an in vitro reporter assay (Fig. 2h) that indicated that the presence of the Foxg1-targeted enhancer-like region (PBS1) suppresses the activity of the *COUP-TFI* promoter (Fig. 2h). To further confirm Foxg1-mediated COUP-TFI regulation in vivo, we introduced two gRNAs (g1386 and g1889) with Cas9 to remove the 503-bp PBS1 region in the E13.75 cortex using IUE and analyzed the embryos at E15.5 (Fig. 2i–l). In contrast to the loss of COUP-TFI expression in conditions of Foxg1 GOF alone (Fig. 2a–c), COUP-TFI expression was maintained under Foxg1

overexpression conditions when the Foxg1-targeted region was removed by the two sgRNAs compared with either of the sgRNAs introduced alone (Fig. 2k, l and Supplementary Fig. 1). Taken together, these results demonstrate that Foxg1 suppresses COUP-TFI expression in the developing neocortex through direct binding to the PBS1 enhancer.

**COUP-TFI directs layer 4 fate**. We next investigated the consequence of this mutual expression in the specification of neocortical laminar subtypes. Based on our examination, L4 cells

**Fig. 2** Foxg1 suppresses COUP-TFI through an H3K4me1-enriched enhancer. **a** Schematic diagram of in utero electroporation. **b** COUP-TFI (red) and GFP (green) immunohistochemistry of E14.5 cortices. Closed arrowheads indicate GFP cells with COUP-TFI expression, open arrowheads indicate GFP cells without COUP-TFI expression. Dashed lines indicate the ventricular surface. **c** Quantitative analysis of the percentage (±SEM) of GFP cells that express COUP-TFI. Brains were introduced with *pCAG:GFP* and *pCAG:empty* (Control) or *pCAG:GFP* and *pCAG:Foxg1* (Foxg1 GOF). **d** Schematic diagram showing the strategy of conditional Foxg1 loss-of-function (Foxg1 cKO) using *Emx1^Cre/+* mice. **e** Foxg1 (red), GFP (green), COUP-TFI (red), and Hoechst 33342 (blue) staining of E14.5 cortices. Dashed lines indicate the ventricular surface. **f** Schematic view of the *COUP-TFI* gene loci (chr13:78,316,360-78,342,718, UCSC genome browser, NCBI37/mm9). The data represent two independent ChIP-seq analyses (Foxg1-ChIP-1 and Foxg1-ChIP-2 from Kumamoto et al.[20]). Mammal Cons represents Placental Mammal Basewise Conservation by PhyloP. MACS peaks overlapping in the two ChIP-seq analyses are marked in gray boxes. Two putative binding sites, PBS1 and PBS2, and primers designed for ChIP-qPCR analysis are indicated (see Methods for details). **g** ChIP-qPCR analyses of PBS1 and PBS2 of cells isolated from E14.5 neocortex using anti-Foxg1 (TaKaRa) and anti-H3K4me1 (Abcam) antibodies. For negative controls, *Foxg1^LacZ/LacZ* (Foxg1 KO) neocortex (Foxg1-ChIP analysis) and IgG (H3K4me1-ChIP analysis) were used. **h** Quantitative *COUP-TFI* promoter activity (±SEM) in the presence of PBS1 by in vitro reporter analysis in U87 human glioblastoma cell line. Value indicates RLU compared to COUPTFIpro (COUP-TFI promoter) only. **i** Schematic diagram showing two gRNAs, g1386 and g1889, designed to target PBS1. **j** Schematic diagram of in utero electroporation. **k** Foxg1 (blue), GFP (green), and COUP-TFI (red) staining of E15.5 cortices. Closed arrowheads indicate GFP cells with COUP-TFI expression and open arrowheads indicate GFP cells without COUP-TFI expression. Brains were introduced with *pCAG:GFP* and *pCAG:empty* (Control) or *pCAG:GFP* and *pCAG:Foxg1* (Foxg1 GOF). Dashed lines indicate the ventricular surface. **l** Quantitative analysis of the percentage (±SEM) of GFP cells that express COUP-TFI. * indicates *P* value <0.05 by Student's *t*-test. Source data are provided as a Source Data file

express low Foxg1 but high COUP-TFI (Fig. 1j–j″), and Foxg1-suppresses COUP-TFI expression (Fig. 2). Therefore, we hypothesized that COUP-TFI expression is necessary for cells to acquire L4 fate. Because COUP-TFI is known to regulate cortical regionalization[29–32], we attempted to manipulate COUP-TFI expression without attenuating cortical area establishment. To this end, we introduced the CRISPR/Cas9 system to disrupt the *COUP-TFI* gene together with the GFP reporter to label future L4 cells using IUE (Fig. 3a). Two pairs of gRNAs targeted to the first or second exon of the *COUP-TFI* gene were cloned into constructs containing the *Cas9* gene (Fig. 3a). These constructs were introduced into the E13.75 mouse neocortex by IUE to target future L4 cells (Fig. 3b). Two days after the introduction, *COUP-TFI* expression in GFP-expressing cells was downregulated in the presence of gRNAs (Fig. 3c). Quantification of COUP-TFI-expressing cells among GFP-positive cells showed 76.8% and 71.2% reduction at E15.5 (Fig. 3d) and 87.9% and 92.8% reduction at P7 (Supplementary Fig. 2) with the introduction of g14 + g216 and g864 + g890, respectively, indicating efficient disruption of the *COUP-TFI* gene using the CRISPR/Cas9 system.

We next analyzed the cell-autonomous requirement of COUP-TFI on L4 cell fate in the primary somatosensory cortex at P7, the timing of which positioning and thalamocortical connectivity of L4 cells has been established (Fig. 3e–u). In the control cortex, the majority of GFP-positive cells were located in somatosensory barrels with high Rorβ expression and showed spiny stellate morphology (Fig. 3e–e″, h, m, n, q). However, COUP-TFI-deficient cells were excluded from the barrels and extended a thick apical dendrite, resembling a pyramidal-like morphology (Fig. 3f–h, m). These GFP-positive cells extended long-distance callosal axons through the corpus callosum and projected to the contralateral side (Fig. 3j–l). The majority of *COUP-TFI* knockout cells lost Rorβ expression and in turn gained Brn2 and Satb2 expression (Fig. 3n–u). A proportion of *COUP-TFI* knockout cells lost upper-layer neuron identity as indicated by loss of Cux1 expression; however, there was no increase in the expression of Ctip2, a marker of deep-layer long-range SCPNs (Fig. 3u). Together, the morphological and molecular identity indicated the primary and cell-autonomous role of COUP-TFI in controlling L4 fate, which secondarily affects sensory area specification[33].

**Foxg1 attenuates layer 4 fate**. Loss of COUP-TFI impairs L4 fate acquisition (Fig. 3), and Foxg1 directly binds to the *COUP-TFI* gene to suppress its expression (Fig. 2f–l); thus, the low Foxg1 expression in L4 cells (Fig. 1j–j″) implies that Foxg1 may

negatively regulate the fate of L4 neurons through COUP-TFI repression. We thus introduced the *Foxg1* gene with the *CAG* promoter together with the GFP reporter into E13.75 progenitor cells (Fig. 4a). Examination at P7 showed control cells positioned in somatosensory barrels with spiny stellate morphology (Fig. 4b, b′, d, d′, e). In contrast, cells that overexpressed Foxg1 as indicated by GFP-positive cells showed pyramidal cell morphology with apical dendrites (Fig. 4c, c′, d, d′) and changes in cell positioning (Fig. 4e). These neurons extended long-range projections through the corpus callosum (Supplementary Fig. 5c, c′, d), which was rarely observed in control GFP neurons (Supplementary Fig. 5b, b′, d). Quantitative analysis showed that *CAG:Foxg1*-overexpressing cells lost Rorβ expression and gained Brn2 and Satb2 expression (Fig. 4g, g′, j, j′, m, m′, o), as opposed to control cells that exhibited a Rorβ^high, Brn2^low, and Satb2^low pattern (Fig. 4f, f′, i, i′, l, l′, o). In addition, conditional Foxg1 overexpression using the *NeuroD1* promoter also changed cell morphology (Fig. 4d′) and molecular expression (Fig. 4h, h′, k, k′, n, n′, o), and the position of *NeuroD1*:Foxg1-overexpressing cells shifted upward (Fig. 4e, *P* = 0.01 in BIN3). These results suggested that ectopic Foxg1 expression hampers L4 fate acquisition.

We next asked whether Foxg1 segregates L2/3 and L4 projection neurons upstream of Brn2 and Rorβ, which exhibit reciprocal expression after E18.5[23]. Quantitative analysis showed that at E18.5, Foxg1-overexpressing cells showed a significant increase in Brn2 expression (Supplementary Fig. 6i–k), indicating that Foxg1 regulates cell identity upstream of this transcriptional network. Notably, Foxg1-expressing cells were distributed in deeper parts of the CP at this stage (Supplementary Fig. 6g, h).

Previously, Foxg1 was found to promote cortical progenitor cells to enter the cell cycle[34], as well as control the migration of cortical neurons after exiting the cell cycle[35]. Therefore, changes in laminar identity upon Foxg1 overexpression may be the consequence of cell cycle re-entry or delayed migration. We thus examined the expression of the proliferation markers Ki67 and pH3 1 day after Foxg1 IUE at E14.5. There was no difference in the number of Ki67- or pH3-positive cells between control and Foxg1-overexpressing cells (Supplementary Fig. 6a–e′), implying that the effect of Foxg1 overexpression on L4 fate is not due to cell cycle re-entry. We next assessed whether Foxg1 alters cell fate through attenuating cell migration by examining the positions of GFP-positive cells. As mentioned above, at E18.5, the majority of control cells arrived above the Ctip2/Zfpm2-expressing deep-layer cells; however, many Foxg1-overexpressing cells were migrating through Ctip2/Zfpm2 deep-layer cells (Supplementary Fig. 6f, g, h, n, o), raising two possibilities: Foxg1 regulates cell

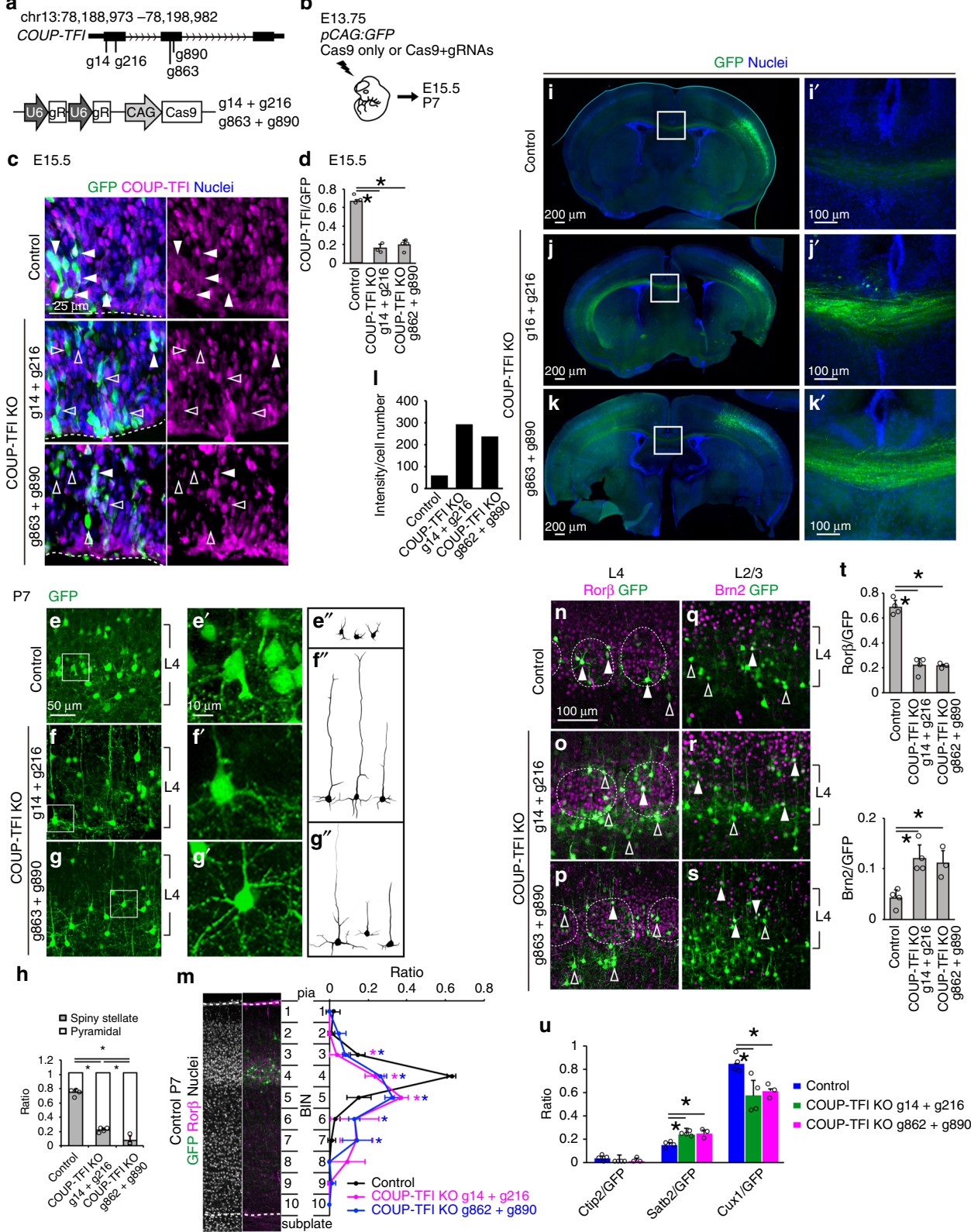

migration, and the altered positioning changed the molecular identity of these cells. Alternatively, Foxg1 defines cell fate, and cells are sorted according to their molecular identity. To differentiate between these possibilities, we tested the consequence of Foxg1 overexpression in deep-layer cells and assessed whether these cells also affect migration and convert to a more upper-layer fate (i.e., L4 cells). We introduced Foxg1 into deep-layer progenitor cells at E12.5 (Fig. 4p). At P7, cells with Foxg1 overexpression were retained in deep layers showing pyramidal morphology similar to that of control cells (Fig. 4q), without increasing the number of Rorβ-positive L4 cells (Fig. 4r–v). Notably, a delayed migration pattern was still observed at 3 days

**Fig. 3** COUP-TFI is required for layer 4 fate acquisition. **a** Schematic diagram showing four gRNAs designed to target the *COUP-TFI* gene. gRNAs, g14 and g216, target the first exon of the *COUP-TFI* gene; gRNAs, g863 and g890, target the second exon of the *COUP-TFI* gene. **b** Schematic diagram of in utero electroporation. **c** GFP (green), COUP-TFI (red) immunohistochemistry, and Hoechst 33342 (blue) of E15.5 cortices. Brains were introduced with *pCAG:GFP* and UbC:Cas9 (Control) or *pCAG:GFP*, UbC:Cas9 and gRNAs (COUP-TF1 KO). gRNA combinations are as indicated. Dashed lines indicate the ventricular surface. Closed arrowheads indicate GFP cells with COUP-TFI expression and open arrowheads indicate GFP cells without COUP-TFI expression. **d** Quantitative analysis of the percentage (±SEM) of GFP cells that express COUP-TFI. **e–g″** GFP immunostaining (green) and dendritic reconstruction of GFP cells in P7 cortices. **h** Quantitative analysis of the percentage (±SEM) of GFP cells with spiny stellate or pyramidal morphology. **i–k′** Immunostaining of GFP (green) and Hoechst 33342 (blue) of P7 cortices. **i′–k′** demarcate enlarged view of boxed regions shown in (**i–k**). **l** Quantitative analysis of the GFP signal intensity in the corpus callosum normalized to GFP-positive cell number. **m** Quantitative analysis of the distribution of GFP cells in P7 cortices. Cortical plate is divided into 10 BINs from the pia to the subplate. **n–s** Immunostaining of P7 cortices for GFP with Rorβ (**n–p**) or Brn2 (**q–s**). Dashed lines indicate layer 4 barrel structures. Closed arrowheads indicate GFP cells with Rorβ or Brn2 expression and open arrowheads indicate GFP cells without Rorβ or Brn2 expression. **t, u** Quantitative analysis of the percentage (±SEM) of GFP cells that express Rorβ and Brn2 (**t**), Ctip2, Satb2 or Cux1 (**u**). * indicates *P* value <0.05 by Student's *t*-test. Source data are provided as a Source Data file

after IUE (Supplementary Fig. 6p–s), while cell fate was not affected (Fig. 4r–v). These results indicated that Foxg1 directly suppresses the fate of L4 cells, thereby causing laminar distribution changes.

**FOXG1 binds to COUP-TFI gene locus to suppress layer 4 fate**. The effect of Foxg1 overexpression on L4 cells coincided with that of the loss of COUP-TFI in these cells (Fig. 3), suggesting a reciprocal regulation of Foxg1 and COUP-TFI to control L4 fate. We thus asked whether COUP-TFI loss-of-function causes Foxg1 upregulation to attenuate L4 fate or if COUP-TFI is a critical Foxg1 downstream regulator that derepresses L4 fate. Knockout of *COUP-TFI* using CRISPR/Cas9 did not affect Foxg1 expression 2 days after electroporation (Supplementary Fig. 4a–d), indicating that Foxg1 upregulation was not the primary cause of the loss of L4 identity upon COUP-TFI deficiency. In contrast, Foxg1 overexpression caused rapid downregulation of COUP-TFI in these cells (Figs. 1d and 2b, c). These results suggested that COUP-TFI is a critical downstream target of Foxg1 that promotes L4 fate.

As we identified a Foxg1-targeted site on the *COUP-TFI* gene (Fig. 2f–l), we next asked whether elimination of this binding of Foxg1 to COUP-TFI could rescue the loss of L4 fate upon Foxg1 overexpression. We introduced the CRISPR system to disrupt the Foxg1-targeted site together with Foxg1 overexpression into future L4 cells by IUE at E13.75 (Fig. 5a). At P7, cells without Foxg1 overexpression were positioned in L4 and developed spiny stellate morphology (Fig. 5b, c). These cells expressed typical L4 markers, Rorβ and Cux1 (Fig. 5b, c, f, g, n, q), were in contact with vGlut2-positive thalamic axon terminals (Fig. 5f, g), and did not express Brn2, Satb2, or Ctip2 (Fig. 5b, c, j, k, o, p). Notably, while disruption of the Foxg1-targeted site alone had no significant effect on L4 molecular identity, the position of these cells shifted slightly towards the lower part of L4. In contrast, in line with previous observations, cells with Foxg1 overexpression alone were distributed away from L4 and exhibited pyramidal morphology (Fig. 5d), accompanied by a loss of Rorβ and Cux1 expression and thalamocortical input (vGlut2) (Fig. 5d, h), but gained Brn2 and Satb2 expression (Fig. 5h, l, o, p). Compared to the introduction of single gRNA, the introduction of two gRNAs rescued the number of L4 cells, indicating that these cells were refractory to Foxg1 overexpression (Fig. 5e, i, m–q). Cells with two gRNAs and Foxg1 expression were repositioned to L4 and showed spiny stellate morphology (Fig. 5e), showing a L4 molecular identity with Rorβ and Cux1 but not Brn2 or Satb2 expression (Fig. 5n–q). Taken together, these results identify COUP-TFI as a critical downstream target of Foxg1 that regulates L4 cell fate acquisition.

**COUP-TFI promotes layer 4 fate acquisition**. Based on the results above, we hypothesized that the expression of COUP-TFI alone may be sufficient to establish L4 cell fate. We thus introduced COUP-TFI into non-L4 precursor cells and directly assessed whether those cells acquire L4 cell identity. First, we introduced COUP-TFI with GFP at E13.25 to target future L5a cells. At postnatal stages, the majority of control GFP cells were found in L5a and did not express the L4 marker Rorβ or L5b marker Ctip2 but expressed Satb2 (Supplementary Fig. 7a–d)[36,37]. Thus, we concluded that E13.25 electroporation mainly targets L5a CPNs. We next examined the effect of COUP-TFI GOF in these future L5a cells (Fig. 6a). Consistent with the results in L4 cells, COUP-TFI overexpression did not affect Foxg1 expression 2 days after its introduction (Fig. 6b). At P8, control cells positioned below L4 not contacting vGlut2 thalamocortical axons did not express Rorβ or Cux1 (Fig. 6c, d, f, h, j, l, n). In contrast, COUP-TFI overexpression increased the number of cells integrated into L4 barrel structures. These cells exhibited spiny stellate-like morphology and expressed L4 markers (Fig. 6c, e, g, i, k, m, n), and an ectopic barrel was occasionally detected (Fig. 6m, m′, compared with the contralateral side in Fig. 6m″). Notably, a few cells outside of barrel structures expressed L4 markers Rorβ and Cux1 and were enriched with vGlut2 signals (Supplementary Fig. 7g, i). These results indicated that COUP-TFI can promote future L5a cells to acquire L4 cell fate.

We further tested whether COUP-TFI overexpression could convert L2/3 cells to L4 cells by introducing COUP-TFI at E15.25 (Fig. 6o and Supplementary Fig. 7j–l). At P8, control cells were positioned in L2/3 and expressed Brn2 and Satb2, and devoid of contact with vGlut2-positive thalamic axon terminals (Fig. 6p, q, s, u, w, y). Cells with COUP-TFI overexpression were shifted deeper, and some of them were positioned in L4 and acquired spiny stellate morphology (Fig. 6p). These cells lost the L2/3 molecular identity of Brn2 and Satb2 expression, and a proportion of them expressed Rorβ and received vGlut2-thalamic axon contacts (Fig. 6r, t, v, x, y). Taken together, these results indicate that COUP-TFI GOF can override the intrinsic genetic program to promote L4 fate.

**Downregulation of Foxg1 is required for layer 4 competence**. Although COUP-TFI overexpression could reprogram non-L4 cells to acquire L4 fate, the efficiency of this reprogramming was higher in L5a cells (Fig. 6a–n) than in L2/3 cells (Fig. 6o–y), implying the influence of other factors. Our results showed that Foxg1 expression in L5a cells was lower than that in L2/3 cells (Fig. 1j–j″) and that Foxg1 suppresses COUP-TFI (Fig. 2) and L4 fate (Fig. 4); however, COUP-TFI manipulation did not affect Foxg1 expression (Fig. 6b and Supplementary Fig. 4a–d). These observations suggest that Foxg1 downregulation may be a

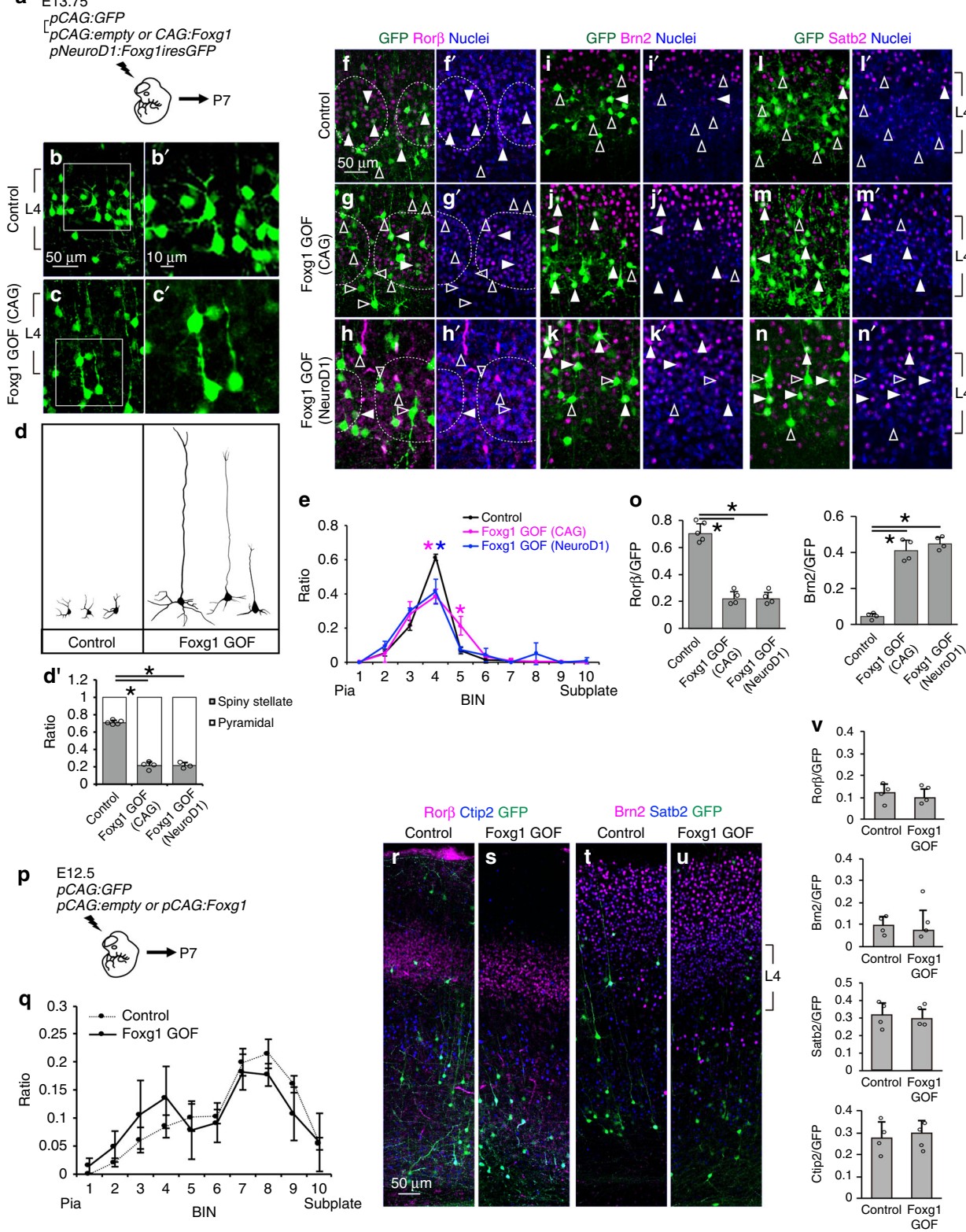

prerequisite to establish L4 cell competence. As Foxg1 have roles in progenitor proliferation[34], we introduced an inducible in vivo loss-of-function approach using the pInducer tet-inducible knockdown system (Fig. 7a). Two shFoxg1 were designed to knockdown Foxg1 expression, and the efficiency was confirmed by qRT-PCR analysis (Supplementary Fig. 8a, b). These constructs were introduced into the E14.5 neocortex by IUE, and

doxycycline was administered from E16.75, at which point the cells have already left the cell cycle (Fig. 7a and Supplementary Fig. 8c, d). At P7, control cells were positioned above L4 and showed pyramidal morphology (Fig. 7b, c, f). In contrast, Foxg1 knockdown cells exhibited a prominent shift into L4 and acquired spiny stellate morphology (Fig. 7b, d–f), with downregulated Brn2 and upregulated Rorβ expression (Fig. 7g–o). Notably, some cells

**Fig. 4** Ectopic Foxg1 expression suppresses layer 4 fate. **a** Schematic diagram of in utero electroporation. Brains were introduced with *pCAG:GFP* and *pCAG:* empty (Control) or *pCAG:GFP* and *pCAG:Foxg1* (Foxg1 GOF (CAG)) or *pNeuroD1:Foxg1iresGFP* (Foxg1 GOF (NeuroD1)). **b–d′** Immunostaining (**b–c′**), reconstruction (**d**), and quantitative analysis of the morphology (**d′**) of GFP cells in P7 cortices. **e** Quantitative analysis of the distribution of GFP cells in P7 cortices. Cortical plate is divided into 10 BINs from the pia to the subplate. **f–n′** Immunostaining of P7 cortices for GFP (green) with Rorβ (red) (**f–h′**), Brn2 (red) (**i–k′**), Satb2 (red) (**l–n′**), and Hoechst 33342 (blue). Dashed lines indicate layer 4 barrel structures. Closed arrowheads indicate GFP cells with Rorβ, Brn2, or Satb2 expression and open arrowheads indicate GFP cells without Rorβ, Brn2, or Satb2 expression. **o** Quantitative analysis of the percentage (±SEM) of GFP cells that express Rorβ or Brn2. **p** Schematic diagram of in utero electroporation. Brains were introduced with *pCAG:GFP* and *pCAG:*empty (Control) or *pCAG:GFP* and *pCAG:Foxg1* (Foxg1 GOF). **q** Quantitative analysis of the distribution of GFP cells in P7 cortices. Cortical plate is divided into 10 BINs from the pia to the subplate. **r–u** Immunostaining of P7 cortices for GFP (green) with Rorβ (red) and Ctip2 (blue) staining (**r**, **s**) or Brn2 (red) and Satb2 (blue) staining (**t**, **u**). **v** Quantitative analysis of the percentage (±SEM) of GFP cells that express Rorβ, Brn2, Ctip2, or Satb2. * indicates *P* value <0.05 by Student's *t*-test. Source data are provided as a Source Data file

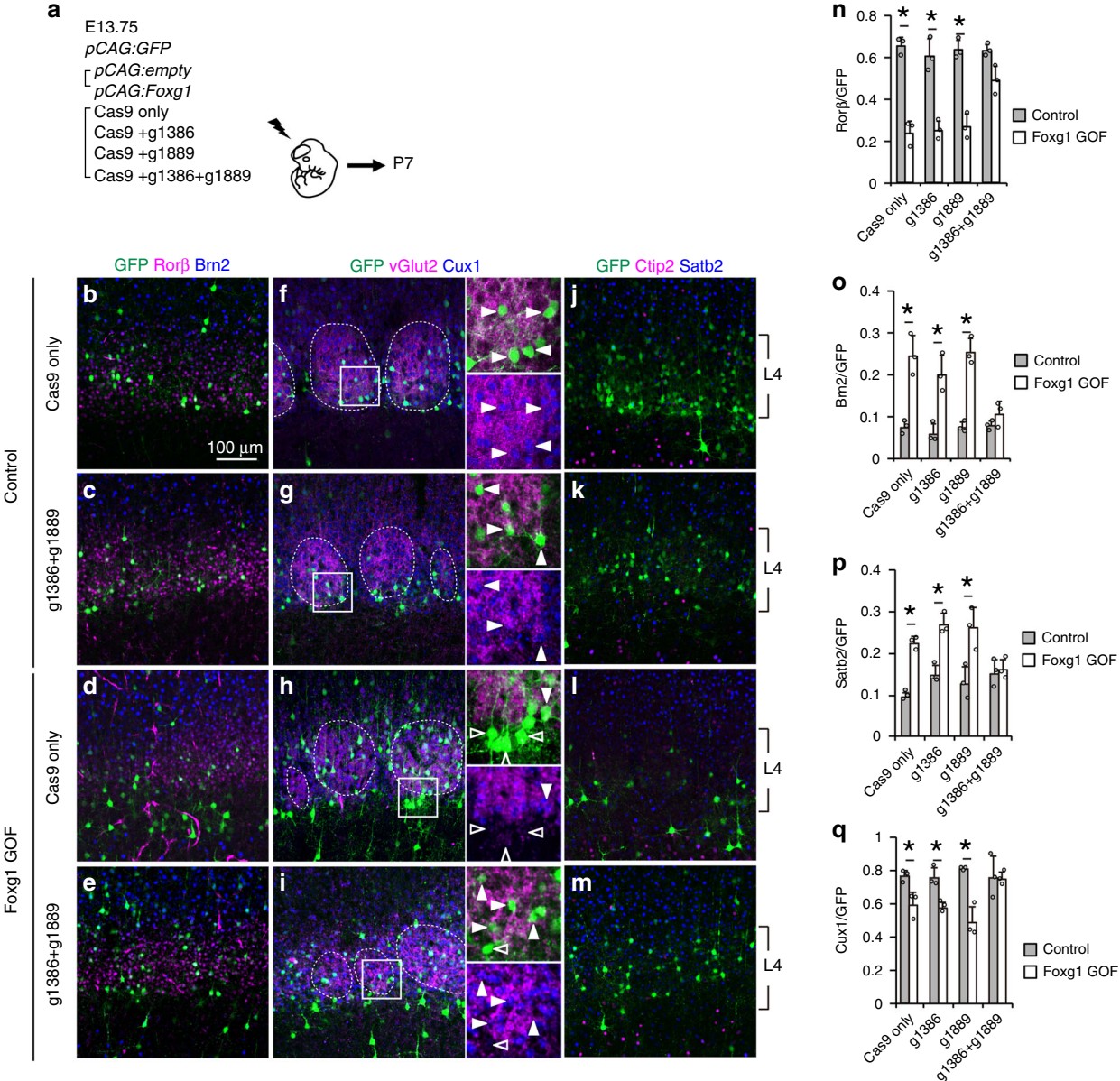

**Fig. 5** Restoration of layer 4 fate upon removal of Foxg1 binding to *COUP-TFI* gene. **a** Schematic diagram of in utero electroporation. Brains were introduced with *pCAG:GFP* and *pCAG:*empty (Control) or *pCAG:GFP* and *pCAG:Foxg1* (Foxg1 GOF). Combinations of gRNAs are as indicated. **b–m** Immunostaining of P7 cortices for GFP (green) with Rorβ (red) and Brn2 (blue) (**b–e**), or vGlut2 (red) and Cux1 (blue) (**f–i**), or Ctip2 (red) and Satb2 (blue) (**j–m**). Dashed lines indicate layer 4 barrel structures. Closed arrowheads indicate GFP cells with enriched vGlut2 signal, thalamocortical presynaptic terminals and open arrowheads indicate GFP cells without enriched vGlut2 signal (**f–i**). **n–q** Quantitative analysis of the percentage (±SEM) of GFP cells that express Rorβ (**n**), Brn2 (**o**), Satb2 (**p**), or Cux1 (**q**). * indicates *P* value <0.05 by Student's *t*-test. Source data are provided as a Source Data file

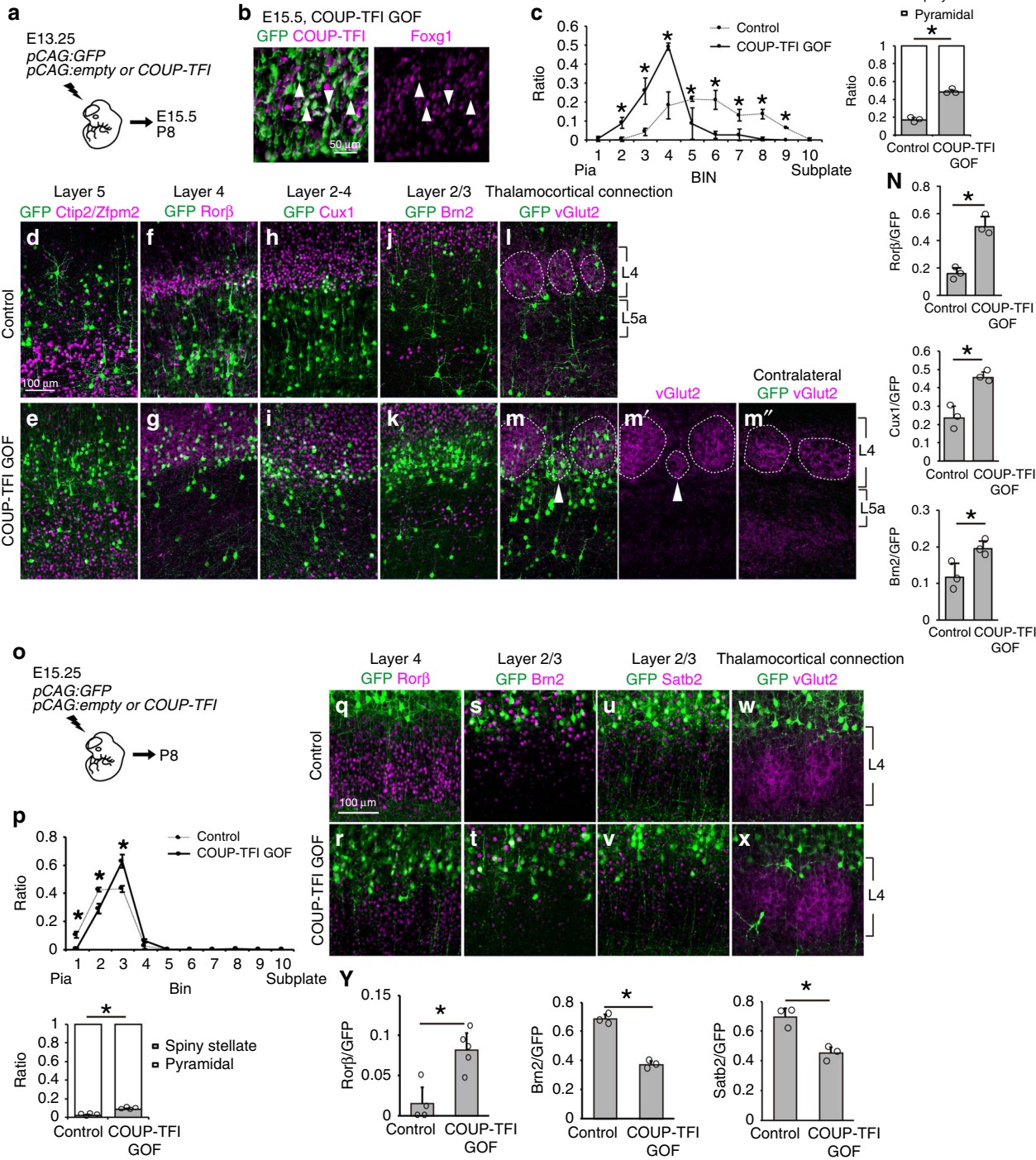

**Fig. 6** Ectopic COUP-TFI expression promotes non-layer 4 cells to acquire layer 4 fate. **a** Schematic diagram of in utero electroporation. Brains were introduced with *pCAG:GFP* and *pCAG*:empty (Control) or *pCAG:GFP* and *pCAG:COUP-TFI* (COUP-TFI GOF). **b** Immunostaining of GFP (green), COUP-TFI (red), and Foxg1 (magenta) in E15.5 cortices. Closed arrowheads indicate GFP cells expressing both COUP-TFI and Foxg1. **c** Quantitative analysis of the distribution of GFP cells in P7 cortices. Cortical plate is divided into 10 BINs from the pia to the subplate. Quantitative analysis of the percentage (±SEM) of GFP cells with spiny stellate or pyramidal morphology. **d-m"** Double immunohistochemistry of P8 cortices for GFP (green) with Ctip2/Zfpm2 (red) (**d**, **e**), Rorβ (red) (**f**, **g**), Cux1 (red) (**h**, **i**), Brn2 (red) (**j**, **k**), and vGlut2 (red) (**l-m"**). Dashed lines indicate layer 4 barrel structures. Closed arrowheads indicate ectopic vGlut2-positive thalamocortical presynaptic terminals as compared to the contralateral hemisphere. **n** Quantitative analysis of the percentage (±SEM) of GFP cells that express Rorβ, Cux1 or Brn2. **o** Schematic diagram of in utero electroporation. Brains were introduced with *pCAG:GFP* and *pCAG*:empty (Control) or *pCAG:GFP* and *pCAG:COUP-TFI* (COUP-TFI GOF). **p** Quantitative analysis of the distribution of GFP cells in P7 cortices. Cortical plate is divided into 10 BINs from the pia to the subplate. Quantitative analysis of the percentage (±SEM) of GFP cells with spiny stellate or pyramidal morphology. **q-x** Double immunohistochemistry for GFP (green) with Rorβ (red) (**q**, **r**), Brn2 (red) (**s**, **t**), Satb2 (red) (**u**, **v**), and vGlut2 (red) (**w**, **x**). **y** Quantitative analysis of the percentage (±SEM) of GFP cells that express Rorβ, Brn2 or Satb2. * indicates *P* value <0.05 by Student's *t*-test. Source data are provided as a Source Data file

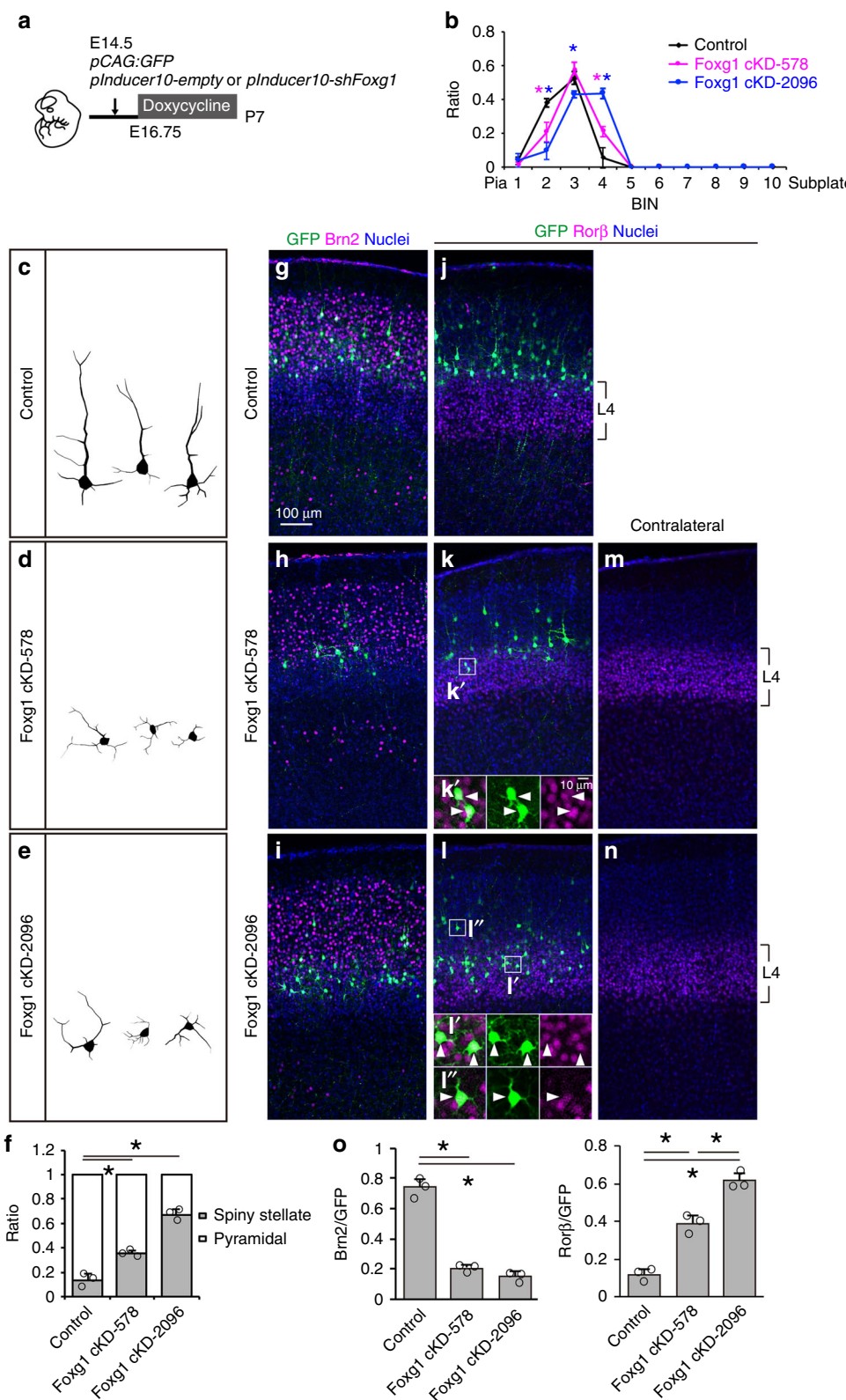

**Fig. 7** Conditional suppression of *Foxg1* converts layer 2/3 to layer 4 neuron identity. **a** Schematic diagram of inducible *Foxg1* knockdown using the *pInducer* system. Doxycycline was administered from E16.75 to suppress *Foxg1* expression. **b** Quantitative analysis of the distribution of GFP cells in P7 cortices. Cortical plate is divided into 10 BINs from the pia to the subplate. **c–e** Representative reconstruction of GFP cells from P7 cortices. **f** Quantitative analysis of the percentage (±SEM) of GFP cells with spiny stellate or pyramidal morphology. **g–n** Double immunohistochemistry for GFP (green) with Brn2 (red) (**g–i**) or Rorβ (red) (**j–n**). **o** Quantitative analysis of the percentage (±SEM) of GFP cells that express Brn2 or Rorβ. * indicates *P* value <0.05 by Student's *t*-test. Source data are provided as a Source Data file

outside of barrel structures displayed spiny stellate dendritic morphology and expressed the Rorβ L4 marker (Fig. 7l″). Collectively, these results demonstrate that downregulation of Foxg1 expression is a key requirement to acquire L4 competence.

**Egr2 directly suppresses Foxg1 to promote layer 4 fate.** Finally, to investigate the upstream molecular mechanism responsible for Foxg1 downregulation in L4 precursor cells, we performed temporal transcriptome analysis to identify genes differentially expressed between earliest L4 and SCPN cells. Previous reports indicated that Neurog2 expression is temporarily upregulated in cells committed to exiting the cell cycle[38]; therefore, we established inducible genetic labeling of temporal precursors by crossing $Neurog2^{CreER/+}$ drivers with Rosa26$^{LSL-tdTomato}$ mice. Upon tamoxifen administration, Cre protein in Neurog2-expressing cells induces recombination at the Rosa26 locus, enabling to label the temporal cohorts of cortical neuron precursors with tdTomato (Fig. 8a). 4OHT administration at E13.5 labeled cell population restricted to deep-layer neurons, while administration at E15.25 selectively labeled L4 cells (Fig. 8b and Supplementary Fig. 10). To examine the difference between L4 cells and deep-layer cell transcripts, tdTomato-positive cells from the somatosensory cortex were sorted using a fluorescence-activated cell sorter (FACS) at 24 and 48 h after 4OHT administration and subjected to RNA sequencing (Fig. 8b). Significant differential expression genes among samples (P value <0.05) showed early differences in deep-layer precursors and L4 precursors (Fig. 8c). Notably, the deep-layer projection neuron-specific genes Ctip2, Fezf2, Tbr1, Sox5, and Foxg1 were enriched in E13.5-labeled cells, whereas the L4 marker gene Pcdh20 was enriched in L4 precursor cells (Fig. 8c). Using this platform, we performed in silico analysis to identify upstream signals. Ingenuity Pathway Analysis found that TGFβ and Fgf signaling were more activated in L4 precursor cells than in deep-layer precursor cells (Fig. 8d). Notably, activated TGFβ and Fgf signaling were detected at both 24 and 48 h after 4OHT administration. To investigate the direct regulator of Foxg1, we submitted 1.3 kb upstream of the Foxg1 gene to TFBIND software (http://tfbind.hgc.jp) to search for TF binding sites (Fig. 8e, f). Apart from common transcriptional regulators, a panel of TF binding sites were found on the Foxg1 upstream region. Interestingly, we identified multiple binding sites of the Egr family within 200 bp (Fig. 8f). Egr1, a previously reported activity-dependent gene[39], was expressed in cortical neurons (Fig. 8g, g′). Egr2, a downstream target of TGFβ signaling[40], was also detected in the intermediate zone and CP (Fig. 8h, h′), and was upregulated in early L4 precursor cells (Fig. 8c). Together, these results suggested that Egr genes control Foxg1 expression to regulate L4 cell competence. To directly assess this hypothesis, we introduced Egr1 or Egr2 into the E14.5 neocortex by IUE (Fig. 8i). At E16.5, Egr2 GOF reduced Foxg1 expression, whereas Egr1 had a milder impact on Foxg1 expression (Fig. 8j–m). At P7, in controls, cells contributed to L2/3 (Fig. 8n–p), and Egr1 overexpression had no effect on L4 molecular identity (Fig. 8q–s, w, x). In contrast, Egr2 overexpression caused a similar effect as conditional Foxg1 knockdown (Fig. 7), where cells shifted to L4 and acquired a Rorβ-positive L4 molecular identity (Fig. 8t–x).

We further assessed the Egr regulation on L4 fate using loss-of-function studies (Fig. 9a–l). We introduced the CRISPR/Cas9 system to disrupt either the Egr1 or Egr2 gene in future L4 cells using IUE at E13.75, and analyzed the consequence at P7 (Fig. 9a–l). Loss of either Egr1 or Egr2 expression caused failure of cells to gain L4 fate and cells exhibited a prominent shift in position (Fig. 9b), acquired pyramidal morphology, lost Rorβ expression, and gained Brn2 expression (Fig. 9d–l). Notably,

Egr1-deficient cells apparently shifted towards deeper regions, while Egr2-deficient cells were excluded from barrels (Fig. 9b), similar to COUP-TFI-deficient cells (Fig. 3) and Foxg1-overexpressing cells (Fig. 4a–o). Furthermore, some Egr1-deficient cells gained Ctip2 expression, which was not observed in Egr2-deficient cells (Fig. 9l). Together with the Egr1 and Egr2 GOF studies (Fig. 8i–x), these results suggest that Egr1 and Egr2 have differential roles in L4 development.

As Egr is predicted to target the Foxg1 promoter to suppress its expression, we further examined whether the presumptive Egr-targeted Foxg1 promoter region is responsible for Egr-Foxg1 regulation in L4 fate selection (Figs. 8e and 9m). We disrupted the Egr-targeted Foxg1 promoter region in future L4 cells using CRISPR/Cas9 (Fig. 9m–v). At E15.5, prominent Foxg1 upregulation in cells introduced with gRNAs were observed, indicating that this region is capable of controlling Foxg1 expression (Fig. 9n). At P7, cells with gRNAs were excluded from L4 and showed pyramidal morphology (Fig. 9o, o′, q, s, u) and expressed less Rorβ (Fig. 9q, q′, v) and more Satb2 (Fig. 9s, s′, v) compared to control cells. The effects of disrupting the Egr-targeted Foxg1 promoter region on attenuating L4 fate were similar to those of disrupting Egr2 expression (Fig. 9a–l), Foxg1 overexpression (Fig. 4a–o) and COUP-TFI loss-of-function (Fig. 3). Conversely, Egr2 overexpression (Fig. 8i–x), conditional Foxg1 disruption (Fig. 7) and COUP-TFI overexpression studies (Fig. 6) resulted in L4 fate acquisition. Taken together, these results identify an Egr-Foxg1-COUP-TFI regulatory network in the preselection of projection neuron subtypes (Fig. 10).

## Discussion

L4 neurons play fundamental roles in the cerebral cortex, acting as a gateway for peripheral inputs to initiate sensory information processing in higher order brain regions[12]. Such a unique function has evolved increased susceptibility of L4 cells to extrinsic cues than distally projecting CFuPNs and CPNs. Therefore, their developmental mechanisms has required the consideration of both cell-autonomous and non-cell autonomous mechanisms. We provide lines of evidence that demonstrate that L4 cell competence is established through an Egr-Foxg1-COUP-TFI network to confer their unique molecular and hodological properties.

In the neocortex, the tight correlation between cell fate and laminar identity[41,42] has raised a question concerning the causal relationship between cell positioning and fate specification of projection neuron subtypes. In this study, removal of COUP-TFI or the acquisition of Foxg1 in L4 precursors resulted in delayed cell migration (Supplementary Figs. 4e–h and 6f–h); however, these neurons were shifted either superficially or deeper relative to the somatosensory barrels and lost L4 molecular identity (Figs. 3 and 4a–o). These results indicate that the fate switch is the primary cause underlying laminar identity changes and not the converse. In support of this notion, our results show that Foxg1 overexpression in deep-layer cells does not result in these cells acquiring a L4 cell fate (Fig. 4p–v), consistent with the aberrant lamination but correct molecular specification seen in the Reln mutant cortex[9].

Currently, the control of the timing of sequential projection subtype specification is still under debate[14,16,17,43,44]. Our study suggests that the fate of L4 neurons is determined relatively early during development. As Foxg1 GOF using the CAG or NeuroD1 promoter blocked L4 fate acquisition (Fig. 4a–o) and COUP-TFI GOF by the NeuroD1 promoter failed to promote L4 fate (Supplementary Fig. 9), the time window in which the fate of L4 neurons is specified appears to be restricted. Based on the position of E14.5-targeted cells at the timing of doxycycline treatment

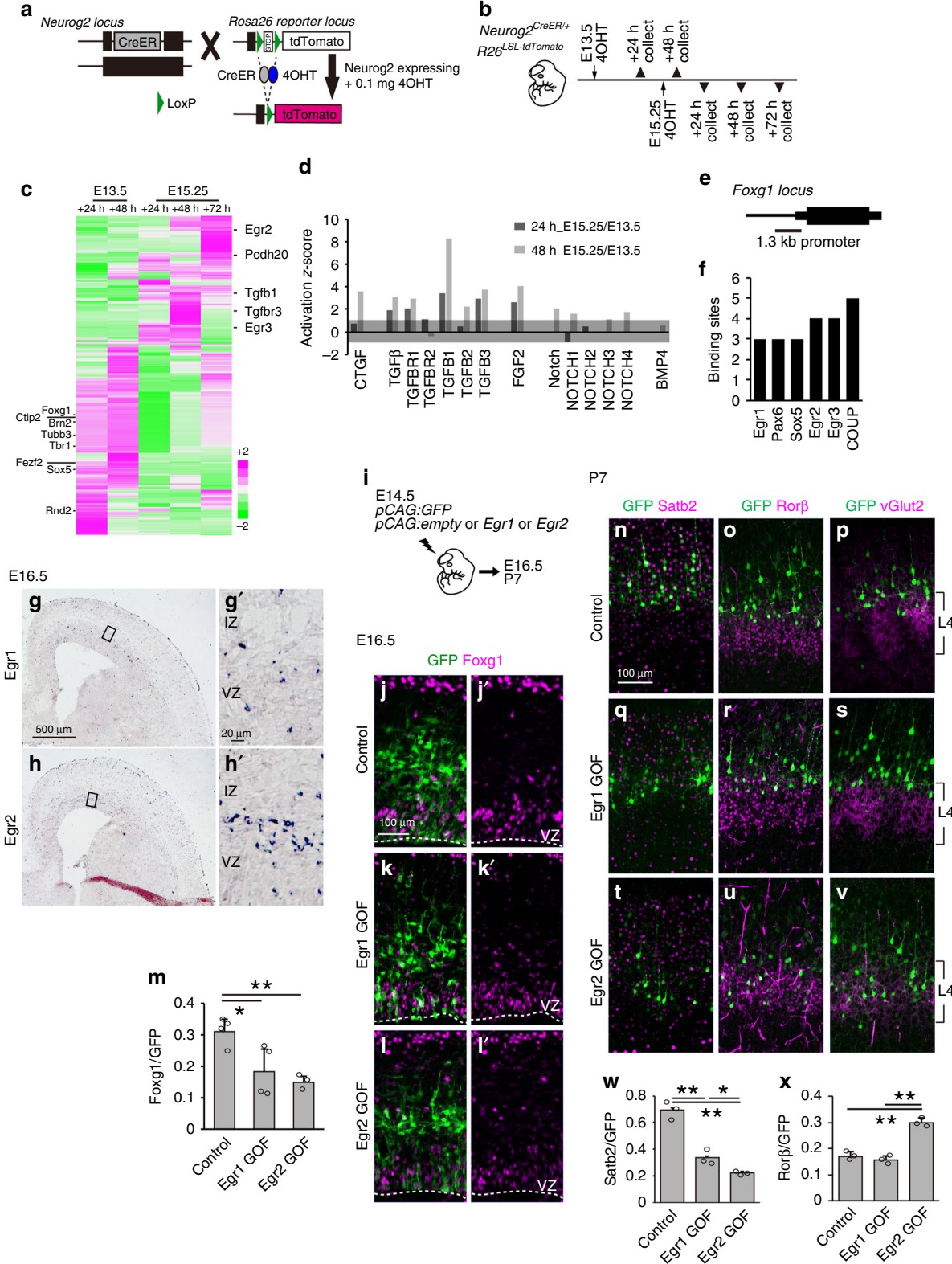

(i.e., *Foxg1* conditional knockdown, Supplementary Fig. 8c, d) and the spatiotemporal expression dynamics of *Egr2* (Fig. 8h, h′), Foxg1 and COUP-TFI (Fig. 1e–e″), together with the reported NeuroD1 expression[45], Neurog2 expression[46], and transcriptomic profile during neurogenesis (Supplementary Fig. 11)[47], the timing of high-Egr2 and low-Foxg1 expression coincides with Tbr2 expression prior to NeuroD1 peak expression (Supplementary

Fig. 11). This Egr2-Foxg1 regulation causes subsequent COUP-TFI upregulation, which may explain why COUP-TFI GOF by the *NeuroD1* promoter is too late to promote L4 fate. Taken together, the results indicate that short- and long-range projection neuron fate commences at the early postmitotic stage, where suppression of Foxg1 by Egr2 results in COUP-TFI upregulation to specify L4 fate. We identified TGFβ signaling as a Foxg1

**Fig. 8** Identification of Foxg1 upstream regulators in projection subtype selection. **a** Schematic diagram of temporal precursor cell labeling using *Neurog2*[CreER/+] and *R26*[LSL-tdTomato] mediated genetic recombination upon 4OHT administration. **b** Schematic diagram of sample collection. *Neurog2*[CreER/+]; *R26*[LSL-tdTomato] mice were administered with 4OHT at E13.5 or E15.25, and brains were collected at the indicated time points. **c** Heatmap representing hierarchical clustering using the complete linkage with Euclidean distance. Datasets were obtained from two independent analyses normalized to *Z*-score and shown in average. Genes with significant *P* value less than 0.01 by differential expression on edgeR and with enriched expression in E13.5 or E15.5 4OHT administration were selected. **d** Ingenuity Pathway Analysis comparing cells from E15.25 and E13.5 4OHT treatment at 24 and 48 h. Candidate upstream regulators related to intercellular signaling pathway with *P* value less than $10^{-3}$ are shown. Positive and negative values of *Z*-score represent activated or repressed state, respectively. Threshold of *Z*-score $= 1$ and $-1$ is marked in gray. **e** Schematic diagram indicating the 1.3 kb *Foxg1* promoter positioned at chr12:50,483,149–50,585,435 on the UCSC genome (NCBI37/mm9). **f** Quantitative analysis of transcription factors with multiple predicted binding sites on the same strand of the *Foxg1* gene (http://tfbind.hgc.jp). **g–h**′ Expression of Egr1 and Egr2 mRNA by in situ hybridization in E16.5 cortices. **i** Schematic diagram of in utero electroporation. Brains were introduced with *pCAG:GFP* and *pCAG:empty* (Control) or *pCAG:GFP* and *pCAG:Egr1* (Egr1 GOF), or *pCAG:GFP* and *pCAG:Egr2* (Egr2 GOF). Dashed lines indicate the ventricular surface. **j–l**′ Double immunohistochemistry of GFP (green) and Foxg1 (red) in E16.5 cortices. **m** Quantitative analysis of the percentage (±SEM) of GFP cells that express Foxg1. **n–v** Immunostaining of P7 cortices with GFP (green), Satb2 (red) (**n, q, t**), Rorβ (red) (**o, r, u**), and vGlut2 (red) (**p, s, v**) antibodies. **w, x** Quantitative analysis of the percentage (±SEM) of GFP cells that express Satb2 (**w**) or Rorβ (**x**). VZ ventricular zone, IZ intermediate zone. * indicates *P* value <0.05 and ** indicates P value <0.01 by Student's *t*-test. Source data are provided as a Source Data file

---

upstream regulator of its repression in L4 cell development (Fig. 8). These results are consistent with a previous reported function of TGFβ in cortical development, where intraventricular TGFβ application suppressed Foxg1 expression in cortical progenitor cells[48], implying a conserved regulation of the TGFβ-Foxg1-COUP-TFI pathway in cortical cell fate specification.

The complementary expression of COUP-TFI and Foxg1 in the developing neocortex is indicative of the reciprocal regulation between these two factors (Figs. 1 and 2). Our results demonstrated that the direct binding of Foxg1 to the COUP-TFI regulatory element is responsible for the suppression of L4 fate (Figs. 3–5), and that Foxg1 downregulation is a prerequisite for L4 cell fate determination upon COUP-TFI GOF (Fig. 7). Previous studies have shown that misexpression of COUP-TFI by the D6 promoter suppresses Fezf2 expression[49] and that Fezf2 expression is derepressed by Foxg1 through suppressing Tbr1[21]. As Fezf2 is the central mediator of SCPN differentiation[22], collectively, these results reveal for the first time the transcriptional regulatory network that segregates distal and local projection neurons mediated through Foxg1-COUP-TFI-Fezf2.

While the conclusions drawn from this study rely on null function experiments, the fine-tuning of projection cell types may also require dose-dependent control of TFs. Indeed, the complementary roles of COUP-TFI and Foxg1 in brain development are consistent with the appearance of the opposed phenotype observed in human pathogenic variants[50–53]. These studies imply that the imbalance between motor outputs and sensory inputs through dysregulated projection subtype selection may lead to altered sensory processing in congenital neurodevelopmental disorders.

## Methods

**Animals.** Two independent conditional *Foxg1* knockout mouse lines were utilized in this study. For the tet-inducible *Foxg1* mouse line, *Foxg1*[tetOFoxg1] conditional mutant mice were generated by crossing *Foxg1*[tTA/+] mice with *Foxg1*[lacZ/+]; tetO*Foxg1*–IRES*lacZ* double-heterozygous mice as previously described[20]. *Foxg1*[tTA/+] heterozygous littermates were used as controls. To control *Foxg1* expression in vivo, 200 μg doxycycline in 250 μL 5% sucrose was administered orally on the 1st day, and 200 mg/L doxycycline in 5% sucrose was provided in drinking water during treatment. For conditional *Foxg1* removal in the dorsal telencephalon (Foxg1 cKO), *Emx*[Cre/+]; *Foxg1*[fl/+]; *R26R*[CAG-FRTstop-EGFP/+] (RCE-FRT) mice were crossed with *Foxg1*[fl/fl]; *RCE-FRT* mice[35,54], and *Emx1*[Cre/+]; *Foxg1*[fl/+]; *RCE-FRT* mice were used as controls. *Foxg1* null mutants (Foxg1 KO) were generated by intercrossing *Foxg1*[lacZ/+] heterozygous mice[55].

To isolate temporal cortical neuronal precursors, we crossed *Neurog2*[CreER/+] heterozygous mice[56] with *R26*[CAG-LSL-tdTomato] mice[57] (Fig. 8a). (Z)-4-Hydroxytamoxifen solution (4OHT, H7904, Sigma) was prepared in 100% ethanol and further diluted with corn oil to reach a final concentration of 1 mg/ml. A total of 0.1 mg/100 μL 4OHT per mouse was administered by intraperitoneal injection. The day of vaginal plug detection was designated as embryonic day (E) 0.5. Both

males and females were used in the experiments. Mice were housed in the Animal Housing Facility of the RIKEN Center for Developmental Biology (Kobe, Japan) and Waseda University Animal Facility (Tokyo, Japan) following the institutional guidelines.

**Constructs.** For *Foxg1* GOF studies, *pCAG:Foxg1* and *pNeuroD1:Foxg1iresGFP* constructs were used as previously described[20,35]. For GOF, *COUP-TFI*, the *Egr1* and *Egr2* genes were amplified using primers (COUP-TFI_forward: GCTCCCTGGGCC CAAAGAT; COUP-TFI_reverse: TCTCCTGGTTTGCAGCTCAG; Egr1_forward: ACCACCCAACATCAGTTCTC; Egr1_reverse: AAGAAAGCAAAGGGAGAGGC; Egr2_forward: TGTGCGAGGAGCAAATGATG; Egr2_reverse: CACCGTGAGATG AAGCTCT) from cDNA prepared from E17.5 mouse neocortex and subcloned into the *pCAGGS* vector backbone (*pCAG:COUP-TFI*, *pCAG:Egr1*, and *pCAG:Egr2*) or the *pNeuroD1:iresGFP* backbone (*pNeuroD1:COUP-TFIiresGFP*). To disrupt the Foxg1-targeted region (PBS1 KO, Figs. 2i–l and 5), *COUP-TFI* gene (COUP-TF1 KO, Fig. 3), *Egr1* gene (Egr1 KO, Fig. 9a–l), *Egr2* gene (Egr2 KO, Fig. 9a–l), or Egr-targeted region (Fig. 9m–v), the CRISPR/Cas9 system was applied[58] using the following designed sgRNAs (http://chopchop.cbu.uib.no)[59]: two sgRNAs for PBS1 KO: g1386: TCTACGCGGCGTACTTGCCTCGG, g1889: TCACCAGCTTCGGAAACA TCGGG; two pairs of sgRNAs for COUP-TFI KO: g14: TGCTGGCTCTGGCCTG AACCGGG, g216: TAGCAGCTGGCGAGATCCGCAGG for the first exon and g863: GGTCCATGAAGGCCACGACGCGG, g890: CCACCTGTTCCTGAAAGAT GCGG for the second exon; two sgRNAs for Egr1: GGCGATCGCAGGACTCGA-CAGGG and TGGCTTCCCGTCGCCGTCAGTGG; two sgRNAs for Egr2: ATCCGTAATTTTACTCTGGGGGGG and GGAGCGAAGCTACTCGGATACGG; and two sgRNAs for the Egr-targeted region: g61: GTAGCCGCGATCGATCATC CTGG, g352: TTCACAGCCGAGCTCGCCGCGGG. sgRNA was cloned into the pX330 backbone according to the instruction manual (Addgene). For conditional Foxg1 knockdown experiments (Foxg1 cKD, Fig. 7), inducible shRNA constructs were named according to the position of the shRNA targets: Foxg1 cKD-578 targeting the coding sequence: CCTGACGCTCAATGGCATCTA; Foxg1 cKD-2096 targeting the 3′ UTR: GCCTTCAGTTTGTGTTGTGTA. Hairpin oligonucleotides were annealed in vitro using annealing buffer containing 10 mM Tris–HCl pH 7.5, 50 mM NaCl and 1 mM EDTA with 50 μM forward and reverse oligonucleotides. The following thermal cycle was used: 95 ℃ for 2 min, followed by cooling to 20 ℃ by decreasing the temperature by 5 ℃/10 min. Annealed oligonucleotides were inserted into the EcoRI- and XhoI-digested pInducer10-mir-RUP-PheS vector (Addgene).

**In utero electroporation.** DNA solutions were prepared with 0.1% Fast Green (Wako) in Hanks' balanced salt solution (HBSS) at the following concentrations: for Foxg1 GOF, 0.5 μg/μL *pCAG:GFP* and 1 μg/μL *pCAG:empty* or *pCAG:Foxg1*; for Foxg1 GOF by *NeuroD1* promoter, 1 μg/μL *pNeuroD1:Foxg1iresGFP*; for COUP-TFI GOF, 1 μg/μL *pCAG:GFP* and 1 μg/μL *pCAG:empty* or *pCAG:COUP-TFI*; for COUP-TFI GOF by *NeuroD1* promoter, 1 μg/μL *pNeuroD1:COUP-TFIiresGFP*; for COUP-TFI KO, PBS1 KO, Egr1 KO, Egr2 KO, and Egr-targeted region KO, 1 μg/μL *pCAG:GFP* and 1 μg/μL *pUbC:Cas9* or *pUbC:Cas9* with gRNAs; for Foxg1 GOF with PBS1 KO: 1 μg/μL *pCAG:GFP*, 1 μg/μL *pCAG:empty* or *pCAG:Foxg1*, and 1 μg/μL *pUbC:Cas9* or *pUbC:Cas9* with gRNAs. Pregnant wild-type ICR mice were deeply anesthetized by intraperitoneal injection using Nembutal sodium solution (Lundbeck) or three types of mixed anesthetic agents containing 5.0 mg/kg butorphanol (Meiji Seika), 4.0 mg/kg midazolam (SAN-DOZ), and 0.75 mg/kg medetomidine (NIPPON ZENYAKU KOGYO CO.)[60]. After uterine horn exposure, the DNA solution was transuterally injected into the lateral ventricles with a 0.6-mm inner-diameter capillary (GD-1, NAR-ISHIGE) pulled by the PC-100 capillary puller (NARISHIGE). The embryo stage

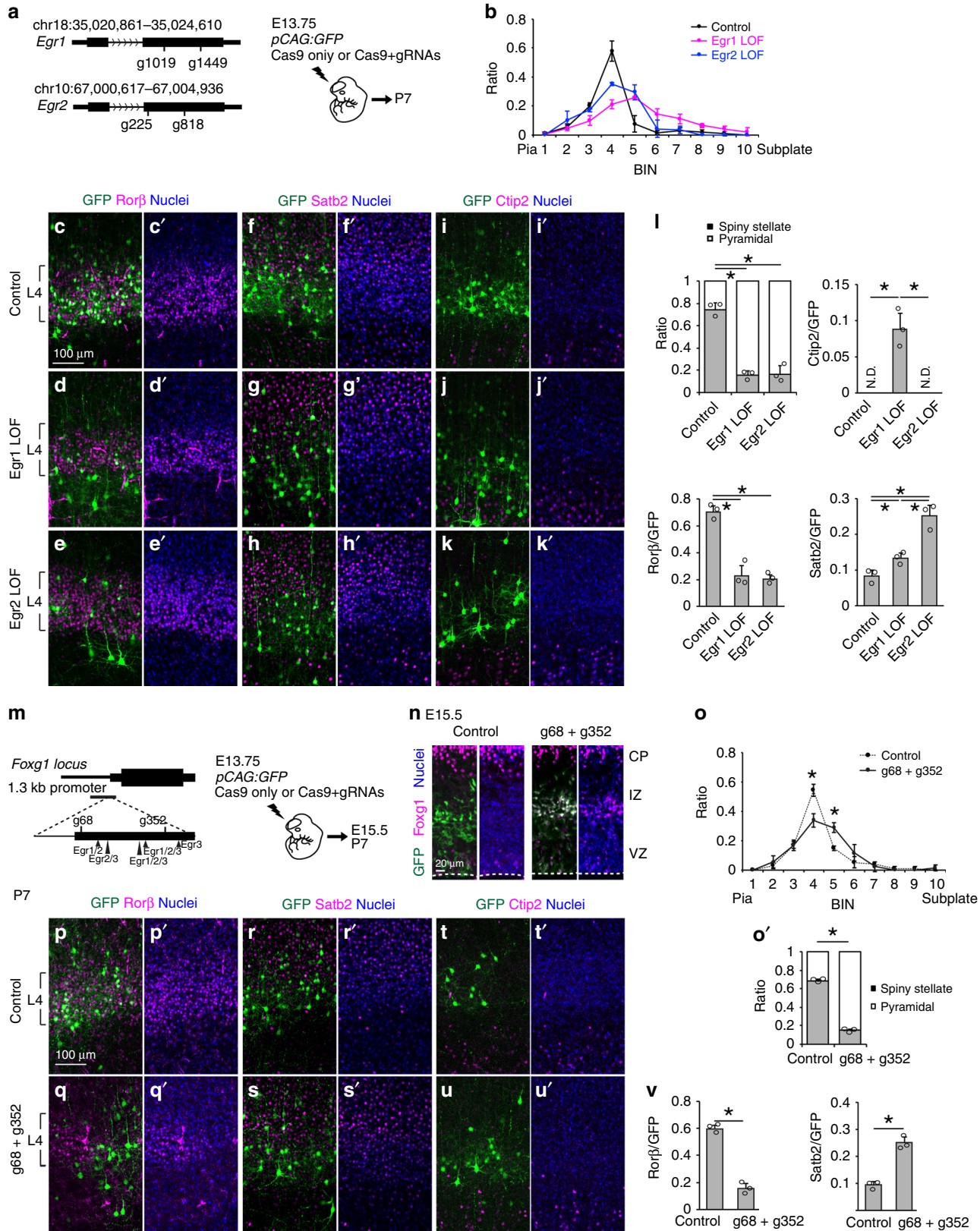

and corresponding voltage were as follows: E12.5 and E13.25: 30 V; E13.75: 33 V; E13.75 and E14.5: 35 V, with a 50-ms on/950-ms off period with a total of 4 pulses using a platinum electrode (CUY650P5; Nepagene). Primary somato-sensory area was targeted throughout this study. After electroporation, the abdominal wall and skin were sutured, and the mice were kept at 37 °C until recovery.

**Immunohistochemistry and image acquisition**. Embryonic brains up to E17.5 were collected directly, and brains from embryos >E17.5 and postnatal stages were perfused with PBS and 4% paraformaldehyde (PFA) prior to fixation. Embryonic brains were fixed with 4% PFA at 4 °C for 1 h, and postnatal brains were fixed with 2% PFA overnight. For cryosection, brains were immersed in 30% sucrose–PBS at 4 °C overnight and embedded in Tissue-Tek O.C.T compound (Sakura). 12-µm-

**Fig. 9** Removal of Egr target site derepresses Foxg1 and suppresses layer 4 fate. **a** Schematic diagram of gRNA design and in utero electroporation. **b** Quantitative analysis of the distribution of GFP cells in P7 cortices. Cortical plate is divided into 10 BINs from the pia to the subplate. **c–k′** Immunohistochemistry of P7 cortices using GFP (green) and Rorβ (red) (**c–e′**), Satb2 (red) (**f–h′**), Ctip2 (red) (**i–k′**) antibodies and Hoechst 33342 (blue). **l** Quantitative analysis of the percentage (±SEM) of GFP cells with spiny stellate or pyramidal morphology and the percentage (±SEM) of GFP cells that express Rorβ or Satb2, or Ctip2. **m** Schematic diagram of gRNA design and in utero electroporation. Brains were introduced with *pCAG:GFP* and Cas9 only (Control) or *pCAG:GFP* and Cas9 with gRNAs (g68 + g352). **n** Immunostaining of E15.5 cortices for GFP (green) with Foxg1 (red) staining and Hoechst 33342 (blue). Dashed lines indicate the ventricular surface. VZ ventricular zone, IZ intermediate zone, CP cortical plate. **o**, **o′** Quantitative analysis of the distribution of GFP cells in P7 cortices. Cortical plate is divided into 10 BINs from the pia to the subplate (**o**). Quantitative analysis of the percentage (±SEM) of GFP cells with spiny stellate or pyramidal morphology (**o′**). **p–u′** Immunostaining of P7 cortices with GFP (green) and Rorβ (red) (**p–q′**), Satb2 (red) (**r–s′**), Ctip2 (red) (**t–u′**) antibodies and Hoechst 33342 (blue). **v** Quantitative analysis of the percentage (±SEM) of GFP cells that express Rorβ or Satb2. * indicates *P* value <0.05 by Student's *t*-test. Source data are provided as a Source Data file

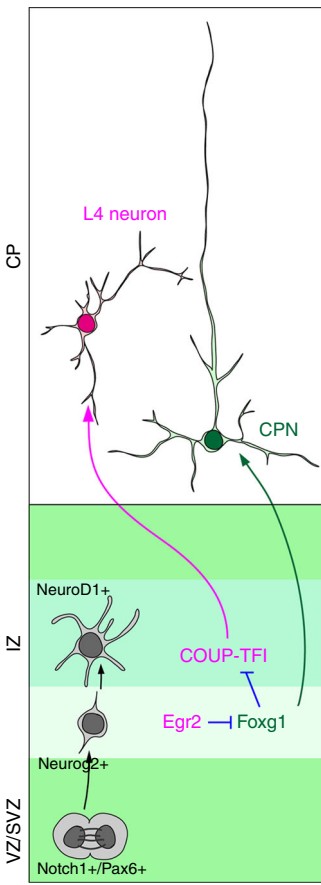

**Fig. 10** Proposed model for short- and long-range projection neuron selection. After the cell cycle exit, Egr2 is activated in early postmitotic neurons and suppresses Foxg1 expression in the lower intermediate zone. This downregulation of Foxg1 derepresses COUP-TFI in postmitotic neurons and confers short-range projection neuron identity. VZ/SVZ ventricular zone/subventricular zone, IZ intermediate zone, CP cortical plate

thick sections were collected using a cryostat (Microm HM550; Carl Zeiss) and stored at −80 °C until analysis. For floating sections, postnatal brains were embedded in 1.4% agarose and 1% gelatin in PBS and sectioned using a vibratome slicer (150 μm) (NLS-AT, Dosaka). Floating sections were processed immediately for immunostaining. For immunostaining on cryosections, blocking solution containing 10% normal donkey serum and 0.05% Triton X-100 in PBS was applied at room temperature for 1 h, and primary antibodies were applied in 1% normal donkey serum and 0.05% Triton X-100 in PBS at 4 °C overnight. After the tissue was washed, secondary antibodies were applied in 0.05% Triton X-100 in PBS at room temperature for 30 min. Nuclei were stained with Hoechst 33342 (ThermoFisher). For immunostaining on floating sections, blocking solution containing 10% normal donkey serum, 0.3% Triton X-100, and 0.1% sodium azide in PBS was applied at 4 °C overnight. Primary antibodies were applied in 1% normal donkey serum, 0.3% Triton X-100, and 0.1% sodium azide in PBS at 4 °C for 3 overnights.

After the tissue was washed, secondary antibodies were applied in 1% normal donkey serum, 0.3% Triton X-100, and 0.1% sodium azide in PBS at 4 °C overnight. Nuclei were stained with Hoechst 33342 (ThermoFisher). Sections were mounted with SlowFade Gold Antifade Mountant (ThermoFisher). Images were acquired using an Axio Imager Z1 fluorescence microscope (Carl Zeiss), FV1000 confocal microscope (Olympus), LSM780 and LSM800 confocal microscope (Carl Zeiss). For dendritic reconstruction, the semiautomated tracing software Neurolucida was applied for images captured using an LSM800 confocal microscope. The following primary antibodies were used: rat anti-GFP (1:500, NACALAI), chicken anti-GFP (1:500, Abcam), rabbit anti-BF1/Foxg1 (1:500, TaKaRa), mouse anti-COUP-TFI (1:200, Perseus Proteomics), rat anti-Ctip2 (1:500, Abcam), mouse anti-Rorβ (1:100, Perseus Proteomics), rabbit anti-Cux1/CDP (1:100, Santa Cruz), goat anti-Brn2 (1:100, Santa Cruz), rabbit anti-Satb2 (1:500, Abcam), rabbit anti-Zfpm2 (1:100, Santa Cruz), guinea pig anti-vGlut2 (1:500, Millipore), rabbit anti-DsRed (1:500, Clontech), mouse anti-Ki67 (1:100, BD), and rabbit anti-pH3 (1:200, Millipore). Secondary antibodies conjugated with Alexa Fluor dye were used at 1:500 dilutions (ThermoFisher), except donkey anti-chicken-488 was used at a 1:200 dilution (Jackson ImmunoResearch).

**In situ hybridization.** Digoxigenin (DIG)-labeled Foxg1 antisense probes were prepared as previously described[61]. *Egr1* and *Egr2* antisense probes were prepared from FANTOM clones (I730052D02 for Egr1 and E430002H10 for Egr2) and used as previously described[62]. Briefly, samples were pretreated and hybridized with probes (100 ng/ml) at 55 °C overnight. The anti-DIG antibody (1:500, Roche) was applied at 4 °C overnight. *Foxg1* signals were developed using Texas-Red streptavidin (1:500, Vector) at room temperature for 2 h. After the tissue was washed with PBS, immunostaining was performed from a blocking step with 10% normal donkey serum and 0.05% Triton X-100 in PBS. The primary antibody, mouse anti-COUP-TFI (1:200, Perseus Proteomics), was diluted in 1% normal donkey serum and 0.05% Triton X-100 in PBS and incubated at 4 °C overnight, and an Alexa Fluor 488-conjugated secondary antibody was used. For *Egr1* and *Egr2* probes, signals were developed with nitro blue tetrazolium (NBT) and 5-bromo-4-chloro-3-indolyl-phosphate (BCIP) substrates (Roche) at room temperature. Images were acquired using an Axio Imager Z1 fluorescence microscope (Carl Zeiss).

**Chromatin immunoprecipitation quantitative PCR.** Neocortical cells from E14.5 embryonic *Foxg1^(lacZ/+)* heterozygote or *Foxg1^(lacZ/lacZ)* null brains were dissociated into single cells and fixed with 1% (v/v) formaldehyde for 8 min at room temperature, and the reaction was quenched with 125 mM glycine. For chromatin immunoprecipitation (ChIP) assay, chromatin was sheared into an average size of 200 bp[63]. For each reaction, $2 \times 10^5$ cells with 2 μg antibody were used (rabbit anti-BF1 antibody, TaKaRa; rabbit anti-H3K4me1 antibody, Abcam; control IgG, Abcam). The isolated DNA was purified using a High Pure PCR Produce Purification Kit (Roche) and quantified by qPCR using SYBR™ Green PCR Master Mix and an ABI PRISM© 7900 Sequence Detection System (Applied Biosystems) with E14.5 mouse neocortical genomic DNA as a standard. Primers (Fig. 2f, g) were as follows: PBS1-1F, TGGGAGGGGCAGATAATGGA; PBS1-1R, AGCTTCGGA AACATCGGGTT; PBS1-0F, AGCCGATAATGCATTAGCTCTCA; PBS1-0R, CCGCCGAGTAAAATCGAGGA; PBS1+1 F, GGGGCTGTGCAGGTGTATAT; PBS1+1 R, GGGGCGAGTGAGCAAACATA; PBS2-1F, CCGCACTGAA ACTCTTGTTGG; PBS2-1R, TAGAGCGAGGTCCATGTCCA; PBS2-0F, TTTGTTGGGGGATGGCAGTT; PBS2-0R, ACTTCAACAGTGCCAGCATT; PBS2+1 F, TCCAGTGTTTTTTGCAGTTGC; PBS2+1 R, CTGTCCTGTGC GATGCCAC. The specificity of the PCR product of each primer pair was confirmed by the presence of a single band on 2% agarose gel electrophoresis and a single peak in the dissociation curve. Enrichment (% Input) was calculated as ChIP DNA/input DNA × 100. Each experiment was repeated at least three times with each replicate producing a similar pattern. The results are shown as the mean ± SEM of three qPCR triplicates.

**Quantitative analysis of callosal axons.** To quantify the long-range callosal axons, we first counted the cell number of electroporated cortices using ImageJ

software (Supplementary Fig. 3a–a‴). The GFP signal was extracted (3a′), and the background signal was subtracted using a 50-pixel rolling ball radius (Supplementary Fig. 3a″). After the threshold was set to 50–255, Analyze Particles function was performed with size >5 and circularity >0.1 to automatically count cell number (Supplementary Fig. 3a‴). The intensity of callosal axons in the corpus callosum was measured based on the quantification method using ImageJ software as described previously[64] (Supplementary Fig. 3b–b‴). The GFP signal in the corpus callosum was extracted (Supplementary Fig. 3b), and the background was subtracted using a 5-pixel rolling ball radius (Supplementary Fig. 3b′). As callosal axons from similar cortical areas occupied a similar position in the corpus callosum, paired regions in a similar position were selected in each condition (Supplementary Fig. 3b″). The signal was plotted, and the area was measured as GFP intensity (Supplementary Fig. 3b‴). The data were calculated as GFP intensity/the number of cells.

**Enhancer reporter assay.** For reporter constructs, the 1.2 kb promoter region of the COUP-TFI gene were cloned into pGL4.1 (Promega) with or without the 550 bp Foxg1 binding site PBS1 (Fig. 2h). pRL-TK containing the Renilla luciferase gene was used as an internal standard for transfection efficiency. Reporter plasmid (0.5 µg) and pRL-TK (0.05 µg) were cotransfected into the U87MG human glioblastoma cell line with Lipofetamine 2000 (Invitrogen) following the Lipofectamine 2000 transfection manual. Two days after transfection, cells were harvested and subjected to the Dual-Luciferase Assay System (Promega). Signals were measured using a 1420ARVOsx-1 luminometer (PerkinElmer Life and Analytical Sciences) according to the manufacturer's standard protocol. Relative luciferase units (RLUs) were calculated as the activity of luciferase/Renilla and normalized to that of the cells transfected with the reporter plasmid containing the COUP-TFI promoter.

**Cell isolation and RNA sequencing.** The putative somatosensory area was isolated from the embryonic neocortex generated by crossing $Neurog2^{CreER/+}$ heterozygous mice with $R26^{CAG-LSL-tdTomato}$ mice with the indicated 4OHT administration (Fig. 8b). Cortical neurons were dissociated into single cells using Neuron Dissociation Solution (Wako) following the standard procedure, and then tdTomato-positive cells were purified with a SH800Z Cell Sorter (Sony Biotechnology Inc.). The RNA from isolated cells was extracted using a QIAquick RNeasy Plus Micro Kit (Qiagen). The quantity and quality of RNA were measured by Qubit (Invitrogen) with a Qubit RNA HS Assay Kit and an Agilent 2100 Bioanalyzer with an Agilent Technologies RNA 6000 Pico Kit. Libraries for paired ends were prepared using the Illumina TruSeq Total RNA Sample Prep Kit following the standard procedure. Two biological replicates were analyzed for each condition.

**Statistical analysis.** The results are shown as the mean ± SEM with ≥3 biological replicates. For statistical analysis, Student's $t$ test or two-way ANOVA was performed, and significance was recognized as $P$ value <0.05. Source data are provided as a Source Data file.

**Reporting summary.** Further information on research design is available in the Nature Research Reporting Summary linked to this article.

## Data availability
The RNA sequence data that support the findings of this study have been deposited in NCBI Sequence Read Archive with the accession number PRJNA505194. Quantitative data underlying Figs. 2c, g, h, l, 3d, h, m, t, u, 4d, e, o, q, v, 5n–q, 6c, n, p, y, 7b, f, o, 8m, w, x, 9b, l, v and Supplementary Figs. 2e, 4d, 6h, k, s, 7d, o, r, 8b, 9b, c, j are available in the Source Data file. The data that support the findings of this study are available from the corresponding author upon reasonable request.

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

## Acknowledgements

The authors thank members of the Hanashima Laboratory for valuable discussions and Q. Wu and D. Campbell for critical reading of the manuscript. The authors thank D. Anderson for the *Neurog2^{CreER}* mice, S. Itohara for the *Emx1^{cre/+}* mice, R. Oda for technical assistance, and C. Tanegashima and the technical support staff of the Laboratory for Phyloinformatics at RIKEN BDR for RNA-seq data acquisition. This work was supported by the Grants-in-Aid for Scientific Research on Innovative Areas ("Interplay of Developmental Clock and Extracellular Environment in Brain Formation") JP16H06483 to C.H., JP17H05775 and JP19H04789 to G.M., and Ministry of Education, Culture, Sports, Science and Technology (MEXT) and Japan Society for the Promotion of Science Grants-in-Aid for Scientific Research (KAKENHI) JP16H04798, JP16K14564, JP17F17090 and JP19H03237 to C.H. and JSPS fellowship to P.S.H., and JP17K07102 and JP19H05228 to G.M.

## Author contributions

P.S.H. and C.H. initiated the study, designed experiments, analyzed the data, and wrote the paper. P.S.H. conducted experiments and collected the data. G.M. provided materials and comments on manuscript.

## Additional information

**Competing interests:** The authors declare no competing interests.

