## [Peer Review File · Nature Communications]

Reviewers' Comments:

Reviewer #1:

Remarks to the Author:

In the submitted manuscript, Hou et al. described the role of Foxg1 and Coup-TF transcription factors in the specification of different types of cortical excitatory superficial layer neurons: spiny stellate cells of layer IV and pyramidal neurons. This is an important and not yet resolved issue in the field, as spiny stellate cells are unique in that they are the main input neurons of the neocortex and their differentiation is not fully understood. The two transcription factors are important players in the developing neocortex with multiple functions at different developmental stages. In the current manuscript the authors investigated late developmental roles of both proteins. For both genes the authors provided in vivo data using loss-of-function and gain-of-function approaches. They showed that FoxG1 promotes specification of long-projecting neurons while COUP-TF promotes specification of Rorb β expressing spiny stellate cells. In a convincing set of both in vitro and in vivo experiments they also demonstrated that Coup-TF expression is negatively regulated by FoxG1 and depends on Coup-TF enhancer. Additionally the authors carried out transcriptome analysis of layer IV precursors and identified Egr1 and Egr2 genes as potential regulators of FoxG1 expression. They could also show that overexpressing Egr1 in the developing neocortex increases the number of FoxG1 cells and changes the proportion of Rorb/Brn2 cells.

This is a well designed and conducted study showing for the first time molecular mechanism of cell fate specification and relations between two neocortical neuronal subtypes: callosal projection neurons and local projection neurons. I think it will have important impact in the field of cortical development.

On the other hand I have two major concerns that should be addressed before publication.

First, although some presented experiments indicate that the cell fate choice between spiny stellate cells and pyramidal neurons seems to take place in postmitotic precursors but not in dividing progenitors, the key experiments to show it directly are missing. To discriminate between the two possibilities I suggest to replicate some GOF experiments with postmitotic promoter, NeuroD or doublecortin. I think it is important to show whether Foxg1 and Coup-TF GOF experiments in postmitotic cells can or cannot shift the cell fate. This would clarify the timing of this important cell fate decision. The experiment with conditional inactivation of Foxg1 points towards postmitotic effect, but because of the way it was designed it is still not a direct evidence.

Egr data do look promising but not comprehensive. On one hand both Egr1 and Egr2 are expressed in a small fraction of cortical cells only on the other hand, functional experiments have been performed with gain-of-function approach, that can be source of artifacts. I would suggest doing Egr loss-of-function experiments, since CrisprCas electroporation seems to work very well. Alternative and even more convincing experiment that the authors could consider is deleting Foxg1 upstream region that contains Egr responsive elements.

Minor issues:

-Fig 2b, Coup-TF staining is relatively weak, so it would be good to see it also alone, without strong GFP signal.

-Fig 2k Here, high magnification picture of cells co-expressing Foxg1 and Coup-TF shows only three cells co-expressing both proteins. It would be more convincing to see it in the same format as on 2b. It would be also more convincing to see all conditions quantified on 2l.

-Why are the numbers of co-expressing cells are so different between 2c and 2l (80% versus 50% in the control and 20 versus 10 in case of Foxg1 OE)?

-Fig3c The efficiency of the Coup-TF KD is difficult to interpret, as GFP signal is very strong and might mask Coup-TF staining. I recommend showing individual channels, at least the red one.

i-k, proportion of callosal axons normalized to number of electroporated neurons should be quantified.

Fig4j and i, would be good to have nuclear staining for Satb2 and Brn1, like it is done for Ror β .
Figure 5f,g that is supposed to show the contact of electroporated cells with thalamocortical

innervation by co-staining with vGlut2 is difficult to interpret, high resolution should be used here. Fig 8n,w shows IHCh for Satb2 cells, but counting for Brn2 cells. I suggest showing data for one or both the markers in both figures.

Reviewer #2:

Remarks to the Author:

In this manuscript, Hou et al. investigated the intrinsic molecular program that specifies sensory neuron identity at the early embryonic development stage, in particular the reciprocal coordination between Foxg1 and Coup-TF1/Nr2f1 in specifying layer 2/3 long-range intracortical projection neurons vs. thalamic input layer 4 local projection neurons. The authors found that overexpression of Foxg1 or knocking out of Coup-TF1 attenuates the acquisition of layer 4 neuronal identity and promotes layer 2/3 neuron identity, whereas Coup-TF1 gain-of-function or knocking out of Foxg1 promotes layer 4 neuron identity. Moreover, they found that Egr, the major target of TGFbeta signaling pathway, can partially repress Foxg1 and mediate layer 4 neuron identity acquisition. The authors performed a series of gain and loss-of-functional studies. The findings are interesting and broaden our knowledge about the intrinsic molecular program in specifying cortical neuronal identity at the early developmental stage.

Main comments:

1. The title appears vague and not particularly relevant to the main findings of the study. A more specific title should be used.
2. One of the key issues related to cortical neuron identity specification is whether the key transcription factors function in progenitor cells or postmitotic neurons or both in specifying neuronal identity. For example, Foxg1 is expressed in VZ progenitor cells (e.g. E11.5-13.5) and postmitotic neurons (e.g. P4). Is Foxg1 required in VZ progenitor cells or postmitotic neurons or both in promoting layer 2/3 neuron identity? Similarly, Coup-TF1 is expressed in VZ cells at E13.5 and in postmitotic neurons postnatally. The authors should take this key issue into consideration in designing and interpreting the experiments.
3. In Figure 1g-g'', 1i-i'' and j-j'', it appears that Foxg1 and Coup-TF1 are coexpressed in CP and VZ cells at E13.5, layer 6 neurons at P1, and layer 2/3 and 4 neurons at P4, respectively. This is inconsistent with the complementary expression of Foxg1 and Coup-TF1 that the authors suggested. The authors should comment on this. How about at the later stage? It would be helpful to quantitatively examine the coexpression between Foxg1 and Coup-TF1. How is the expression patterns of Foxg1 and Coup-TF1 related to neuronal morphology development?
4. Previous study suggested that E13.5 labeled Foxg1 GOF cells become temporarily stuck in the lower part of the intermediate zone with a multipolar morphology, which leads to a delayed migration and ultimate acquisition of layer 2/3 instead of layer 4 neuron identity (Miyoshi and Fishell, 2012).

In this study, the authors argued that the primary function of Foxg1 overexpression is to change neuronal identity but not migration. It is based on the observation that overexpression of Foxg1 in deep layer neurons at E12.5 fail to promote upper layer neuron identity. In this context, did Foxg1 overexpression affect neuronal migration? Given that the interpretation by the authors is fundamentally different from the previous study, the authors should carefully analyze the data to strengthen the conclusion.

5. The authors suggest that an intrinsic preselection program like Foxg1/Coup-TF1 enables cells to attract thalamic axons and then acquire spiny stellate neuron identity. Based on this, is the barrel structure expanded into layer 2/3 or layer 5, given that Coup-TF1 GOF or Foxg1 cKO cells are located not only in layer 4 but also in layer 2/3 and layer 5 (Figure 6 and 7)? Do Coup-TF1 GOF or Foxg1 cKO cells located outside layer 4 display typical layer 4 neuron marker expression, morphology, or circuit connection?

Minor comments:

1. In Figure 4c and c', Only two Foxg1 GOF cells in layer 4 were shown. Were most of Foxg1 GOF cells located in layer 2/3 area at P7?

Also in Figure 4f, it seems that most of cells in the barrel structure are spiny stellate-like cells.

2. Page 22, line 3, should "identify" be "identity"?

Reviewer #3:

Remarks to the Author:

In this manuscript, Hou et al. investigate the molecular mechanisms of projection neuron specification in the mouse neocortex. Building on previously published transcriptional profiling, the authors study the roles of Foxg1 and Coup-TF1 in layer 4 cell differentiation. They use in utero electroporation at different time points during embryonic development to demonstrate that Coup-TF1 expression is necessary for layer IV cell fate acquisition. The authors next show that overexpression of Foxg1 suppresses layer IV cell differentiation. Foxg1 function depends on Foxg1 binding to the Coup-TF1 enhancer, suggesting that Coup-TF1 is a direct target of Foxg1. Overexpression of Coup-TF1 in cells in layer II/III and layer V results in the ectopic expression of layer IV cell markers, an effect partially recapitulated by Foxg1 loss of function. Finally, using gene expression analysis and gain of function manipulations, the authors identify the TGF beta / Egr1 signaling pathway as a potential upstream regulator of Foxg1 function.

The work presented in this manuscript comprises a large set of powerful experiments that have the potential to provide important new insights into the specification of neocortex layer IV neurons and, more specifically, into the functions of Foxg1 and Coup-TF1 during cortex development. However, in its current form, important parts of the data are difficult to interpret. Histological analyses and quantification is difficult to validate, and the description of the data lacks necessary detail. Evaluation of the manuscript is further complicated by its loose structural organization and many English mistakes.

I recommend that the authors revise their manuscript to improve the analysis, quantification, and presentation of their data, before publication in Nature Communications.

Main points:

1) The conclusion that Coup-TF1 and Foxg1 exhibit mutual (mutually exclusive?) expression patterns in developing neocortex is not quantified and largely inconsistent with the data presented in Figure 1. The authors should provide a more rigorous description of their histological analyses.

2) In Figure 2c, co-expression of Coup-TF1 and GFP expressed from a control vector is observed in ~80% of neurons, Foxg1 overexpression reduces co-expression to ~25%. In Figure 2l, control conditions result in ~45%, Foxg1 overexpression in ~10% co-expression. Thus, the variability

observed under control conditions is comparable to the effects caused by Foxg1 overexpression. The authors need to explain this point. The actual numbers (mean, SEM, p values etc) should be provided for all data points throughout the manuscript.

The authors should describe the consequences of PBS1 deletion on wild-type layer IV cells, i.e. in the absence of Foxg1 overexpression.

3) The authors describe 'prominent' shifts in the positioning of layer II/III and V cells upon Coup-TF1 overexpression and Foxg1 knock down. These shifts are difficult to evaluate (Figures 6 and 7). The authors should quantify the positioning and morphological characteristics of cells to support their conclusions.

In Figure 7j, the boxed area does not appear to correspond with the high magnification inset.

4) The authors use narrow time windows for electroporation and doxycycline and tamoxifen administration to target distinct subpopulations of cortical progenitors. The authors should provide a quantification of the variability of targeting distinct layers/cell types, which is expected to reflect variability in the developmental stages of electroporated/injected embryos.

In Figure 6 a-j, the authors claim to selectively target layer 5a cells - the histology does not appear to support this claim.

5) The focus of this work is poorly defined. Abstract and introduction focus on intrinsic versus extrinsic regulation of modality-specific neural circuits. The results primarily describe the consequences of Foxg1 and Coup-TF1 gain and loss of function experiments on layer IV cell specification, but does not provide any insight into modality-specific circuit development. In fact, it remains unclear if experiments were consistently performed in somatosensory cortex or also in other cortical areas. Large parts of the discussion are devoted to the timing of cell fate specification and sensory circuit development and disease.

The manuscript would benefit from tightening up its focus. Many typos and grammatical errors make this manuscript difficult to read.

6) Changes in the projection patterns of Coup-TF1 knock-out neurons (Figure 3i-k) should be quantified. What if the effect of Foxg1 overexpression on layer IV cell projection targets?

Minor points:

7) It would be useful if the authors could provide a summary scheme of the genetic interactions that control layer IV cell fate specification.

8) Histological images should be presented and labeled in a consistent manner, to allow for comparison and better orientation. In Figure 2, for example, 4 different magnifications are used, none of them are placed into a broader context to facilitate orientation.

9) Figure 8 provides evidence for a role of TGF beta / Egr1 signaling in controlling Foxg1 expression in layer IV precursor cells. The experiments that support this conclusion are less comprehensive than the experiments to probe the functional interactions between Foxg1 and Coup-TF1, and the results open up several new questions that remain unanswered. I suggest to focus on Foxg1/Coup-TF1 and address TGF beta / Egr1 signaling in more detail elsewhere. FACS experiments are not described in the Methods.

Point-By-Point Response To The Referees' Comments:

Reviewer #1 (Remarks to the Author):

Reviewer comments: In the submitted manuscript, Hou et al. described the role of Foxg1 and Coup-TF transcription factors in the specification of different types of cortical excitatory superficial layer neurons: spiny stellate cells of layer IV and pyramidal neurons. This is an important and not yet resolved issue in the field, as spiny stellate cells are unique in that they are the main input neurons of the neocortex and their differentiation is not fully understood. The two transcription factors are important players in the developing neocortex with multiple functions at different developmental stages. In the current manuscript the authors investigated late developmental roles of both proteins. For both genes the authors provided in vivo data using loss-of-function and gain-of-function approaches. They showed that FoxG1 promotes specification of long-projecting neurons while COUP-TF promotes specification of Rorb β expressing spiny stellate cells. In a convincing set of both in vitro and in vivo experiments they also demonstrated that Coup-TF expression is negatively regulated by FoxG1 and depends on Coup-TF enhancer. Additionally the authors carried out transcriptome analysis of layer IV precursors and identified Egr1 and Egr2 genes as potential regulators of FoxG1 expression. They could also show that overexpressing Egr1 in the developing neocortex increases the number of FoxG1 cells and changes the proportion of Rorb/Brn2 cells. This is a well designed and conducted study showing for the first time molecular mechanism of cell fate specification and relations between two neocortical neuronal subtypes: callosal projection neurons and local projection neurons. I think it will have important impact in the field of cortical development.

Response: We thank Reviewer 1 for the favorable comments.

Reviewer comments: On the other hand I have two major concerns that should be addressed before publication.

First, although some presented experiments indicate that the cell fate choice between spiny stellate cells and pyramidal neurons seems to take place in postmitotic precursors but not in dividing progenitors, the key experiments to show it directly are missing. To discriminate between the two possibilities I suggest to replicate some GOF experiments with postmitotic promoter, NeuroD or doublecortin. I think it is important to show whether Foxg1 and Coup-TF GOF experiments in postmitotic cells can or cannot shift the cell fate. This would clarify the timing of this important cell fate decision. The experiment with conditional inactivation of Foxg1 points towards postmitotic effect, but because of the way it was designed it is still not a direct evidence.

Response: We thank Reviewer 1 for raising this point and providing critical suggestions. According to our results on the conditional Foxg1 removal study (Figures 7) and the expression pattern of the Egr genes (Figure 8g-h'), we postulated that the fate selection between projection neuron subtypes proceeds at postmitotic stage, specifically, at the time when the cells reach the lower part of the intermediate zone and express NeuroD1 (Miyoshi and Fishell, 2012). Therefore, we performed Foxg1 and COUP-TFI gain-of-function (GOF) studies using the NeuroD1 promoter vectors (Miyoshi and Fishell, 2012) and analyzed the cortex at P7, when laminar identity has been established. Based on these postmitotic GOF studies, we found that NeuroD1:Foxg1 GOF cells lost pyramidal morphology and Rorb β expression, and these cells in turn obtained Brn2 expression (new Figure 4d', e, h-o). We also found that the position of NeuroD1:Foxg1 GOF cells were excluded from layer 4 positions and shifted upwards to layer 2/3 (revised Figure 4e, p = 0.01 in BIN3). Together, these results indicated that upregulation of Foxg1 expression during layer 4 generation hampers layer 4 neuron fate acquisition.

As the critical timing and temporal gene cascade driving this fate selection still remains inconclusive, we further tested this hypothesis by performing postmitotic COUP-TFI GOF using the same NeuroD1 promoter (new Figure S9). We introduced NeuroD1:COUP-TFI constructs at E13.25 and analyzed the cortex at P7. To our surprise, NeuroD1:COUP-TFI GOF cells prominently shifted towards upper layer 2/3 or deeper cortical plate, but avoided layer 4 positions (revised Figure S9b, interaction p < 0.0001 by two-way ANOVA). These NeuroD1:COUP-TFI GOF cells neither gained spiny stellate morphology nor Rorb β expression, but retained pyramidal morphology and obtained Brn2 expression (new Figure S9c-j), including those positioned in deeper cortical plate (new Figure S9g, g', j). These NeuroD1:COUP-TFI GOF results

suggested that the precise timing of COUP-TFI upregulation, specifically, prior to *NeuroD1* expression, is critical for the acquisition of layer 4 cell fate.

As *Foxg1* GOF using *CAG* or *NeuroD1* promoter suppressed layer 4 fate, and COUP-TFI GOF by *NeuroD1* promoter failed to promote layer 4 fate, the time window by which *Foxg1* expression is downregulated and COUP-TFI is upregulated becomes limited. In our conditional *Foxg1* knockdown studies, E14.5 targeted cells resided between the ventricular zone and the intermediate zone at the timing of doxycycline treatment (i.e. E16.75) (revised Figure S8c, d), and these *Foxg1* knockdown cells acquired layer 4 cell competence and identity (revised Figure 7). Considering the spatiotemporal expression of *Egr2* (Figure 8h, h'), *Foxg1* and COUP-TFI expression (Figure 1e-e''), together with the reported *NeuroD1* expression (Hevner et al., 2006), *Neurog2* expression (Britz et al., 2006) and transcriptomic profile during neurogenesis (Telley et al., 2016), the timing of high-*Egr2* and low-*Foxg1* expression coincides with *Tbr2* expression prior to *NeuroD1* peak expression (new Figure S11). This *Egr2*-*Foxg1* regulation causes subsequent COUP-TFI upregulation, which may explain why COUP-TFI GOF by the *NeuroD1* promoter is too late to promote layer 4 fate. Taken together, the results indicate that short- and long-range projection neuron fate commences at early postmitotic stage. During this time window, *Egr2* suppresses *Foxg1* expression, which results in COUP-TFI upregulation to specify layer 4 fate. We have included these new data in the revised Figures 4a-o, S9, S11, and revised manuscript (Page 17 Line 3-6; Page 28, Line 11-25).

- Egr data do look promising but not comprehensive. On one hand both Egr1 and Egr2 are expressed in a small fraction of cortical cells only on the other hand, functional experiments have been performed with gain-of-function approach, that can be source of artifacts. I would suggest doing Egr loss-of-function experiments, since CrisprCas electroporation seems to work very well. Alternative and even more convincing experiment that the authors could consider is deleting Foxg1 upstream region that contains Egr responsive elements.

Response: We thank Reviewer 1 for critical suggestions. We have now assessed the cell-autonomous requirement of *Egr* genes in cortical projection neuron subtype specification. To achieve this, we introduced CRISPR/Cas9 system to disrupt either the *Egr1* or *Egr2* gene together with GFP reporters in future layer 4 cells by *in utero* electroporation at E13.75, and analyzed the cell fate at P7 (new Figure 9a-l). In line with other experiments (Figures 3, 4, 5), the majority of GFP-positive cells in the control cortex were located in layer 4 region with spiny stellate morphology, and expressed *Rorb* but not *Satb2*. In contrast, loss of either of *Egr1* or *Egr2* expression caused failure of these cells to adopt layer 4 identity, and were excluded from barrels, showed pyramidal morphology, and lost *Rorb* expression but gained *Satb2* expression (new Figure 9b-l). However, while the effect of *Egr2* LOF was similar to the positional changes observed in COUP-TFI LOF and *Foxg1* GOF studies (revised Figures 3, 4), *Egr1* LOF cells apparently changed their positions towards deeper regions (new Figure 9b). In addition to this observation, some of the *Egr1* LOF cells gained *Ctip2* expression (new Figure 9j, j', l), which was not observed in *Egr2* LOF, COUP-TFI LOF and *Foxg1* GOF studies (revised Figures 3, 4, 9). Together with the *Egr1* and *Egr2* GOF studies (revised Figure 8), these results indicated that *Egr1* and *Egr2* have different roles on layer 4 development, where *Egr2* primarily acts upstream of the *Foxg1*-COUP-TFI regulatory pathway. Based on these results, we propose that *Egr2* suppresses *Foxg1* to maintain COUP-TFI expression and confer layer 4 fate (revised Figures 8i-x, 9a-l).

As the Reviewer suggested, it is of interest to address whether the presumptive *Egr*-targeted *Foxg1* promoter region is indeed responsible for this *Egr*-*Foxg1* regulation in layer 4 fate selection (Figures 8e, 9m). To test this, we have now introduced CRISPR/Cas9 system to disrupt the *Egr*-targeted *Foxg1* promoter region in future layer 4 cells, and analyzed the fate of these neurons (new Figure 9m-v). Two days after electroporation, we found prominent upregulation of *Foxg1* in the cells targeted with gRNAs (new Figure 9n), indicating that this region is responsible for repressing *Foxg1* expression. At P7, control cells without gRNAs were positioned in the somatosensory barrels with spiny stellate morphology (new Figure 9o, o', p, r, t). In contrast, cells introduced with gRNAs were excluded from layer 4 (new Figure 9o, interaction $p < 0.0001$ by two-way ANOVA) and showed pyramidal morphology (new Figure 9o'). These cells with gRNAs lost *Rorb* expression and gained *Satb2* expression compared to control cells (new Figure 9q-v). The effect of disruption of *Egr*-targeted *Foxg1* promoter region in attenuating layer 4 fate was similar to that observed in *Egr2* LOF (new Figure 9a-l), *Foxg1* GOF (revised Figure 4a-o) and COUP-TFI LOF studies (revised Figure 3). Conversely, *Egr2* GOF (revised Figure 8i-x), *Foxg1* LOF (revised Figure 7) and COUP-TFI GOF (revised Figure 6) resulted in layer 4 fate acquisition. Collectively, these results identify a novel

Egr-Foxg1-COUP-TFI regulatory network in the early segregation of cortical short-range projection neuron fate. We have included all of these results in the new Figure 9 and the manuscript (Page 23, Line 21 to Page 25, Line 3).

Minor issues:

-Fig 2b, Coip-TF staining is relatively weak, so it would be good to see it also alone, without strong GFP signal.

Response: We thank Reviewer 1 for this suggestion. We have included COUP-TFI-only channel in the revised Figure 2b.

-Fig 2k Here, high magnification picture of cells co-expressing Foxg1 and Coup-TF shows only three cells co-expressing both proteins. It would be more convincing to see it in the same format as on 2b. It would be also more convincing to see all conditions quantified on 2l.

Response: As suggested, we have added both the low and high magnification immunostaining images of both Control and Foxg1 GOF cells in the presence of g1386+g1889 in the revised Figure 2k and Figure S1.

- Why are the numbers of co-expressing cells are so different between 2c and 2l (80% versus 50% in the control and 20 versus 10 in case of Foxg1 OE)?

Response: The differences between Figure 2c and Figure 2l come from the differences in the timing of sample collection. In Figure 2c, we aimed to evaluate the primary targets of Foxg1, therefore the brains were collected one day after *in utero* electroporation (i.e. E14.5). In Figure 2l, we aimed to assess the effect of removal of Foxg1-targeted region on *COUP-TFI* gene using the CRISPR/Cas9 system. As it requires longer time for gene editing, we harvested brains two days after *in utero* electroporation, E15.5. At E14.5, many of the electroporated cells were still in the ventricular zone (revised Figure 2b), whereas at E15.5, most of these cells left the ventricular zone and resided in the intermediate zone (revised Figure 2k). Based on our observation, COUP-TFI expression in the intermediate zone is lower than that in the ventricular zone (revised Figure 1e-e'', h-h''), which leads to the differences in the proportion of cells that express COUP-TFI between these population (E14.5, revised Figure 2b, c and E15.5, revised Figure 2k, l).

*-Fig3c The efficiency of the Coup-TF KD is difficult to interpret, as GFP signal is very strong and might mask Coup-TF staining. I recommend showing individual channels, at least the red one.
i-k, proportion of callosal axons normalized to number of electroporated neurons should be quantified.*

Response: As Reviewer 1 suggested, we have included the single COUP-TFI channel in revised Figure 3c, which shows removal of COUP-TFI expression in *COUP-TFI* KO GFP-positive cells as compared to Cas9-only Controls (revised Figure 3c).

In order to quantify the callosal axons in Figure 3i-k', we applied the quantification method by Poo and colleagues (Zhou et al., 2013). As described in the revised Materials and methods and in the new Figure S3, we quantified the signals for callosal projections by selecting a paired region of each condition and analyzed the signal intensity using ImageJ software (Zhou et al., 2013). Cell number was counted using Particle Analysis function, and axon signal intensity was then normalized to cell number and shown as a bar chart in the revised Figure 3l (please refer to the revised manuscript Page 10, Line 4-18 in Materials and methods 'Quantitative analysis of callosal axons' section for detailed image processing procedure). Based on this analysis, we identify increased callosal projections in the *COUP-TFI* KO cortex, which further supports the notion that COUP-TFI removal attenuates layer 4 fate and promotes callosal projection neuronal identity (revised Figure 3i-l). We also performed the same quantitative analysis to assess the impact of Foxg1 acquisition on callosal projections. The results showed increased callosal projections in the Foxg1 GOF cortex (revised Figure S5). Together, these results support our finding that Foxg1 and COUP-TFI reciprocally regulates long- and short-range projection identity.

- Fig4j and i, would be good to have nuclear staining for Satb2 and Brn1, like it is done for Rorb.

Response: As recommended by Reviewer 1, we have included separate channels for Satb2 and Brn2 staining in the revised Figure 4i-n'.

- *Figure 5f,g that is supposed to show the contact of electroporated cells with thalamocortical innervation by co-staining with vGlut2 is difficult to interpret, high resolution should be used here.*

Response: As recommended, we have included high resolution images as well as vGlut2-only channel to show the contact of thalamocortical axon terminals and GFP-positive cells in the revised Figure 5f-i (arrowheads).

- *Fig 8n,w shows IHCh for Satb2 cells, but counting for Brn2 cells. I suggest showing data for one or both the markers in both figures.*

Response: We thank Reviewer 1 for this notification and sincerely apologize for the confusion raised by the original Figure 8w. The confusion was caused by mislabeling of the quantification results obtained from Satb2 staining as Brn2 during the Figure preparation processes. The quantification results in the original Figure 8w was from Satb2 staining, which is now correctly labeled in the revised Figure 8w.

Reviewer #2 (Remarks to the Author):

Comments: In this manuscript, Hou et al. investigated the intrinsic molecular program that specifies sensory neuron identity at the early embryonic development stage, in particular the reciprocal coordination between Foxg1 and Coup-TFI/Nr2f1 in specifying layer 2/3 long-range intracortical projection neurons vs. thalamic input layer 4 local projection neurons. The authors found that overexpression of Foxg1 or knocking out of Coup-TFI attenuates the acquisition of layer 4 neuronal identity and promotes layer 2/3 neuron identity, whereas Coup-TFI gain-of-function or knocking out of Foxg1 promotes layer 4 neuron identity. Moreover, they found that Egr, the major target of TGFbeta signaling pathway, can partially repress Foxg1 and mediate layer 4 neuron identity acquisition. The authors performed a series of gain and loss-of-functional studies. The findings are interesting and broaden our knowledge about the intrinsic molecular program in specifying cortical neuronal identity at the early developmental stage.

Main comments:

1. The title appears vague and not particularly relevant to the main findings of the study. A more specific title should be used.

Response: We thank Reviewer 2 for valuable comments and suggestions. In the revised manuscript, we have now provided new data to support the Egr regulation upstream of Foxg1-COUP-TFI on projection subtype segregation (new Figure 9). Therefore, we have modified the title to ‘Sensory cortex wiring relies on the preselection of short- and long-range projection neurons through an Egr-Foxg1-COUP-TFI network’.

2. One of the key issues related to cortical neuron identity specification is whether the key transcription factors function in progenitor cells or postmitotic neurons or both in specifying neuronal identity. For example, Foxg1 is expressed in VZ progenitor cells (e.g. E11.5-13.5) and postmitotic neurons (e.g. P4). Is Foxg1 required in VZ progenitor cells or postmitotic neurons or both in promoting layer 2/3 neuron identity? Similarly, Coup-TFI is expressed in VZ cells at E13.5 and in postmitotic neurons postnatally. The authors should take this key issue into consideration in designing and interpreting the experiments.

Response: We thank Reviewer 2 for raising this important point concerning the timing of cortical neuron fate selection. According to our current study and previous reports (Hanashima et al., 2004; Hanashima et al., 2002; Miyoshi and Fishell, 2012; Toma et al., 2014), Foxg1 expression expands in the dorsal telencephalic progenitors around E9.5 to regulate cell cycle and terminate Cajal Retzius cells production (Hanashima et al., 2002; Kumamoto et al., 2013). Within the cortex, Foxg1 is expressed in the progenitor cells, downregulated in the multipolar cells migrating in the intermediate zone, and then reactivated before entering the cortical plate (revised Figure 1) (Miyoshi and Fishell, 2012). The dynamics of Foxg1 expression have indicated its pleiotropic roles in progenitor cell proliferation, neuronal precursor cell migration and fate determination at multiple stages during corticogenesis. COUP-TFI, in turn, is expressed in progenitor cells in the ventricular zone, neuronal precursors and cortical neurons (Figure 1) (Faedo et al., 2008; Zhou et al., 2001). While it has been documented that COUP-TFI negatively regulates proliferation and regulates arealization (Faedo et al., 2008), the time window by which stage COUP-TFI regulates corticogenesis has remained poorly understood.

In the original manuscript, we demonstrated that the critical timing of preselection of thalamocortical receipt layer 4 neurons and callosal projection neurons commences in the early postmitotic stage (Figures 7, 8), however, as two of the Reviewers (Reviewer 1 and 2) pointed out, this perspective should be addressed directly. To clarify the timing of Foxg1 and COUP-TFI action on fate selection, we now performed conditional gain-of-function (GOF) studies with the *NeuroD1* promoter, as *NeuroD1* is expressed in the intermediate zone (Hevner et al., 2006). Based on the results of postmitotic Foxg1 GOF driven by *NeuroD1* promoter at E13.75, we found *NeuroD1*:Foxg1 GOF cells lost *Rorb* expression and obtained *Brn2* and *Satb2* expression (new Figures 4d', e, h-o), and were excluded from layer 4 positions (Figure 4e, $p = 0.01$ in BIN3 and 4). The effects on cell identity by *CAG*:Foxg1 GOF and *NeuroD1*:Foxg1 suggested that Foxg1 misexpression hampers layer 4 fate specification.

We further tested the timing when layer 4 fate is selected through postmitotic COUP-TFI GOF driven by the *NeuroD1* promoter (new Figure S9). We introduced *NeuroD1*:COUP-TFI constructs at E13.25 and analyzed the cortex at P7. To our surprise, *NeuroD1*:COUP-TFI GOF cells prominently shifted towards upper L2/3 or deeper part of cortical plate but avoided layer 4 position (new Figure S9b, interaction $p < 0.0001$ by two-way ANOVA). *NeuroD1*:COUP-TFI GOF cells at P7 neither changed morphology into spiny stellate cells nor gained Ror β expression, but retained pyramidal morphology (new Figure S9c) and obtained Brn2 expression (Figure S9j, Brn2 $p = 0.00024$) including those positioned in deeper cortical plate (new Figures S9g, g'). These results indicate that layer 4 fate selection commences in postmitotic neurons, prior to the expression of NeuroD1.

As Foxg1 GOF using CAG or *NeuroD1* promoter suppresses layer 4 fate, and COUP-TFI GOF by *NeuroD1* promoter failed to promote layer 4 fate, the time window by which Foxg1 expression is downregulated and COUP-TFI is upregulated becomes limited. In our conditional *Foxg1* knockdown studies, E14.5 targeted cells reside between the ventricular zone and the intermediate zone at the timing of doxycycline treatment (i.e. E16.75) (revised Figure S8c, d), and these *Foxg1* knockdown cells acquired layer 4 competence and cell identity (revised Figure 7). Considering the spatiotemporal expression of Egr2 (Figure 8h and h'), Foxg1 and COUP-TFI expression (Figure 1e-e''), together with the reported NeuroD1 expression (Hevner et al., 2006), Neurog2 expression (Britz et al., 2006) and transcriptomic profile during neurogenesis (Telley et al., 2016), the timing of high-Egr2 and low-Foxg1 expression coincides with Tbr2 expression prior to NeuroD1 peak expression (new Figure S11). This Egr2-Foxg1 regulation causes subsequent COUP-TFI upregulation, which may explain why COUP-TFI GOF by the *NeuroD1* promoter is too late to promote layer 4 fate. Taken together, the results indicate that short- and long-range projection neuron fate commences at early postmitotic stage. During this time window, Egr2 suppresses Foxg1 expression, which results in COUP-TFI upregulation to specify layer 4 fate. We have included these new data in the revised Figures 4a-o, S9 and S11, and revised manuscript (Page 17 Line 3-6; Page 28, Line 11-25).

3. In Figure 1g-g'', 1i-i'' and 1j-j'', it appears that Foxg1 and Coup-TF1 are coexpressed in CP and VZ cells at E13.5, layer 6 neurons at P1, and layer 2/3 and 4 neurons at P4, respectively. This is inconsistent with the complementary expression of Foxg1 and Coup-TF1 that the authors suggested. The authors should comment on this. How about at the later stage? It would be helpful to quantitatively examine the coexpression between Foxg1 and Coup-TF1. How is the expression patterns of Foxg1 and Coup-TF1 related to neuronal morphology development?

Response: We thank Reviewer 2 for raising this concern. In this study, we utilized both *in situ* hybridization and immunohistochemistry using Foxg1 and COUP-TFI antibodies to assess their spatiotemporal expression dynamics in the developing cortex. Our results indicate higher complementarity in the Foxg1 *in situ* and COUP-TFI immunohistochemistry analysis (Figure 1e, e', e'') compared to Foxg1-COUP-TFI double immunohistochemistry (Figure 1h, h', h''), suggesting the perdurance of Foxg1 protein but not active transcription in a subpopulation of postmitotic neurons. This downregulation of Foxg1 was also apparent in our previous study, which utilized 4 different antibodies for Foxg1 (Toma et al. 2014). The current TaKaRa antibody used in this study does not have the high S/N as compared to discontinued StemCulture and Santa Cruz polyclonal antibody products used in the previous study, indicating that some of the immunostaining signals may under or overrepresent the endogenous Foxg1 levels. [REDACTED]

4. Previous study suggested that E13.5 labeled Foxg1 GOF cells become temporarily stuck in the lower part of the intermediate zone with a multipolar morphology, which leads to a delayed migration and ultimate acquisition of layer 2/3 instead of layer4 neuron identity (Miyoshi and Fishell, 2012). In this study, the authors argued that the primary function of Foxg1 overexpression is to change neuronal identity but not migration. It is based on the observation that overexpression of Foxg1 in deep layer neurons at E12.5 fail to promote upper layer neuron identity. In this context, did Foxg1 overexpression affect neuronal migration?

Given that the interpretation by the authors is fundamentally different from the previous study, the authors should carefully analyze the data to strengthen the conclusion.

Response: We thank Reviewer 2 for this consideration and agree that further examination of Foxg1 GOF at E12.5 would help us clarify the primary effect of Foxg1 on migration versus fate selection. We thus performed short-term analysis to assess the migration pattern of E12.5 Foxg1 GOF cells at E15.5. Consistent with the previous report (Miyoshi and Fishell, 2012), GFP-positive cells in the Foxg1 GOF cortex showed delayed migration as compared to GFP-cells in the control cortex (new Figure S6p-s, interaction $p < 0.0001$ by two-way ANOVA) and revised manuscript (Page 18, Line 9-10). This delayed migration pattern was also observed in E13.75 Foxg1 GOF (Figure S6f-h, interaction $p < 0.0001$ by two-way ANOVA). However, neurons born at E12.5 did not acquire layer 4 fate, while those born at E13.75 lost layer 4 cell identity (Figure 4). These results indicate that delayed migration is not the primary cause for cell fate change observed in layer 4 precursors, and that Foxg1 directly regulates projection neuron subtypes.

5. The authors suggest that an intrinsic preselection program like Foxg1/Coup-TFI enables cells to attract thalamic axons and then acquire spiny stellate neuron identity. Based on this, is the barrel structure expanded into layer 2/3 or layer 5, given that Coup-TFI GOF or Foxg1 cKO cells are located not only in layer 4 but also in layer 2/3 and layer 5 (Figure 6 and 7)? Do Coup-TFI GOF or Foxg1 cKO cells located outside layer 4 display typical layer 4 neuron marker expression, morphology, or circuit connection?

Response: We thank Reviewer 2 for comments and agree that intrinsic conversion of cell fate may lead to barrel structure expansion or Ror β -expressing cells outside of barrels that attract thalamocortical axon terminals as previously reported (Jabaudon et al., 2012). In E13.25 COUP-TFI GOF experiments (Figure 6a-n), we found a small population of cells outside the barrel structure that expressed layer 4 markers Ror β and Cux1, and these cells were enriched with vGlut2-positive thalamocortical presynaptic terminals (revised Figure S7e-i). In addition, we also found ectopic Foxg1 knockdown cells outside of barrels that exhibited spiny stellate morphology and Ror β expression (revised Figure 7i''). We have included these observations in the revised manuscript (Page 20, Line 14-16; Page 21, Line 23-24) and revised Figures 7i'' and S7e-i. In addition, while we found that these few cells outside of barrel structures can exhibit layer 4 character, the prominent positional shift observed in Foxg1/COUP-TFI manipulated cells indicate that intrinsic changes in transcriptional profile switches cell fate at early postmitotic stage, which causes these cells to preferentially integrate into the designated laminar position.

Minor comments:

1. In Figure 4c and c', Only two Foxg1 GOF cells in layer 4 were shown. Were most of Foxg1 GOF cells located in layer 2/3 area at P7?

Response: We apologize for this confusion and have replaced with representative figure showing moderate electroporation efficiency in revised Figures 4c and c'. Based on the results, we found that Foxg1 GOF cells were excluded from barrels and positioned to either upper layer 2/3 or deeper layer 5a region. These cells lost layer 4 marker Ror β expression and gained Brn2 and Satb2 expression (Figure 4g, g', j, j', m, m', o). Additionally, we quantified GFP-positive cell distribution in the cortical plate (Figure 4e). The results showed that Foxg1 GOF cells significantly shifted into deeper positions ($p = 0.012$ in BIN5).

Also in Figure 4f, it seems that most of cells in the barrel structure are spiny stellate-like cells.

Response: In response to the Reviewer's comment, as the results in Figure 4f was obtained from 12 μ m-thick Z-stack, we further confirmed the morphology of these GFP-positive cells in the Foxg1 GOF cortex with thicker Z-stack. Indeed, some of the GFP-positive cells in the barrels showed spiny stellate morphology, while most of them extended long apical processes. To strengthen this observation, we have replaced the new Figures 4g and g' to focus on the cells inside the barrels and provided quantification results showing the significant decrease in the proportion of GFP-positive cells with spiny stellate morphology in the Foxg1 GOF cortex (Figure 4d'). Taken together, these results suggest that ectopic Foxg1 overexpression causes future layer 4 cells to be excluded from barrels and extend long apical processes.

2. Page 22, line 3, should “*identify*” be “*identity*”?

Response: We thank the Reviewer for notification, we have corrected in our revised manuscript (revised manuscript Page 23, Line 17).

Reviewer #3 (Remarks to the Author):

In this manuscript, Hou et al. investigate the molecular mechanisms of projection neuron specification in the mouse neocortex. Building on previously published transcriptional profiling, the authors study the roles of Foxg1 and Coup-TF1 in layer 4 cell differentiation. They use in utero electroporation at different time points during embryonic development to demonstrate that Coup-TF1 expression is necessary for layer IV cell fate acquisition. The authors next show that overexpression of Foxg1 suppresses layer IV cell differentiation. Foxg1 function depends on Foxg1 binding to the Coup-TF1 enhancer, suggesting that Coup-TF1 is a direct target of Foxg1. Overexpression of Coup-TF1 in cells in layer II/III and layer V results in the ectopic expression of layer IV cell markers, an effect partially recapitulated by Foxg1 loss of function. Finally, using gene expression analysis and gain of function manipulations, the authors identify the TGF beta / Egr1 signaling pathway as a potential upstream regulator of Foxg1 function.

The work presented in this manuscript comprises a large set of powerful experiments that have the potential to provide important new insights into the specification of neocortex layer IV neurons and, more specifically, into the functions of Foxg1 and Coup-TF1 during cortex development. However, in its current form, important parts of the data are difficult to interpret. Histological analyses and quantification is difficult to validate, and the description of the data lacks necessary detail. Evaluation of the manuscript is further complicated by its loose structural organization and many English mistakes. I recommend that the authors revise their manuscript to improve the analysis, quantification, and presentation of their data, before publication in Nature Communications.

Main points:

1) The conclusion that Coup-TF1 and Foxg1 exhibit mutual (mutually exclusive?) expression patterns in developing neocortex is not quantified and largely inconsistent with the data presented in Figure 1. The authors should provide a more rigorous description of their histological analyses.

We thank Reviewer 3 for raising this concern. In this study, we utilized both *in situ* hybridization and immunohistochemistry using Foxg1 and COUP-TFI antibodies to assess their spatiotemporal expression dynamics in the developing cortex. Our results indicate higher complementarity in the Foxg1 *in situ* and COUP-TFI immunohistochemistry analysis (Figure 1e, e', e'') compared to Foxg1-COUP-TFI double immunohistochemistry (Figure 1h, h', h''), suggesting the perdurance of Foxg1 protein but not active transcription in a subpopulation of postmitotic neurons. This downregulation of Foxg1 was also apparent in our previous study, which utilized 4 different antibodies for Foxg1 (Toma et al., 2014). The current TaKaRa antibody used in this study does not have the high S/N as compared to discontinued StemCulture and Santa Cruz polyclonal antibody products used in the previous study, indicating that some of the immunostaining signals may under or overrepresent the endogenous Foxg1 levels. [REDACTED]

2) In Figure 2c, co-expression of Coup-TF1 and GFP expressed from a control vector is observed in ~80% of neurons, Foxg1 overexpression reduces co-expression to ~25%. In Figure 2I, control conditions result in ~45%, Foxg1 overexpression in ~10% co-expression. Thus, the variability observed under control conditions is comparable to the effects caused by Foxg1 overexpression. The authors need to explain this point. The actual numbers (mean, SEM, p values etc) should be provided for all data points throughout the manuscript.

Response: We thank Reviewer 3 for the comment, and this concern was also raised by Reviewer 1. The differences between Figure 2c and Figure 2I come from the differences in the timing of sample collection. In Figure 2c, we aimed to evaluate the primary targets of Foxg1, therefore the brains were collected one day after *in utero* electroporation (i.e. E14.5). In Figure 2I, we aimed to assess the effect of removal of Foxg1-

targeted region on *COUP-TFI* gene using the CRISPR/Cas9 system. As it requires longer time for gene editing, we harvested brains two days after *in utero* electroporation, E15.5. At E14.5, many of the electroporated cells are still in the ventricular zone (revised Figure 2b), whereas at E15.5, most of these cells have left the ventricular zone and resided in the intermediate zone (revised Figure 2k). Based on our observation, COUP-TFI expression in the intermediate zone is lower than that in the ventricular zone (revised Figure 1e-e'' and h-h''), which leads to the differences in the proportion of cells that express COUP-TFI between these population (E14.5, revised Figure 2b, c and E15.5, revised Figure 2k, l).

Furthermore, in response to Reviewer's comment concerning the statistical value of each experiments, we have now included the information including the actual numbers (mean, SEM, P values, degrees of freedom and F value) in the new Source Data file.

The authors should describe the consequences of PBS1 deletion on wild-type layer IV cells, i.e. in the absence of Foxg1 overexpression.

Response: As recommended by Reviewer 3, we have included the description of the consequence of PBS1 deletion on wildtype layer 4 cells in the revision (Page 19, Line 9-11). We found that while disruption of the Foxg1-targeted site alone has no significant effect on layer 4 molecular identity, the position of these cells shifted slightly towards the lower part of layer 4. As layer 4 cells are normally devoid of Foxg1 expression (Figure 1j''), these results imply that the population affected by the PBS1 deletion are cells that are normally Foxg1-positive COUP-TFI-negative cells outside of the barrels. Therefore, the loss of PBS1 may increase COUP-TFI expression in these cells, resulting in an excess number of COUP-TFI-positive cells that are positionally shifted within the barrel.

3) The authors describe 'prominent' shifts in the positioning of layer II/III and V cells upon Coup-TFI overexpression and Foxg1 knock down. These shifts are difficult to evaluate (Figures 6 and 7). The authors should quantify the positioning and morphological characteristics of cells to support their conclusions.

Response: We thank Reviewer 3 for suggestions. We have now included the quantification results of distribution and morphological analysis of GFP-positive cell in the revised Figures 6c, 7b, f and in the Source Data file.

In Figure 7j, the boxed area does not appear to correspond with the high magnification inset.

Response: We thank the Reviewer for notifying this. We have corrected the position of box in the revised Figure 7j.

4) The authors use narrow time windows for electroporation and doxycycline and tamoxifen administration to target distinct subpopulations of cortical progenitors. The authors should provide a quantification of the variability of targeting distinct layers/cell types, which is expected to reflect variability in the developmental stages of electroporated/injected embryos.

In Figure 6 a-j, the authors claim to selectively target layer 5a cells - the histology does not appear to support this claim.

Response: We thank Reviewer 3 for raising this point. To identify the cell type targeted by *in utero* electroporation at E13.25, we quantified GFP cells that expresses the following cell type specific markers: layer 5b corticospinal projection neuron gene *Ctip2*, layer 4 neuron gene *Rorb*, layer 2/3 gene *Brn2*, callosal projection neuron gene *Satb2*, and layers 2-4 gene *Cux1*, which showed that the majority of E13.25-targeted cells are *Satb2*-positive callosal projection neurons (revised Figure S7a-d). Previous report on the sequential production of long-range projection neurons indicated that the switch of subcortical *Ctip2*-positive projection neurons to *Satb2*-positive callosal projection neurons occurs between E13.0 to E13.5 (Hatanaka et al., 2016). *Satb2*-positive callosal projection neurons residing in lower layers display genetic hallmarks closer to *Satb2*-positive callosal projection neurons in L2/3 (Klingler et al., 2019), indicating that *Satb2* enrichment in E13.25 electroporated neurons represents layer 5a identity of these neurons.

5) *The focus of this work is poorly defined. Abstract and introduction focus on intrinsic versus extrinsic regulation of modality-specific neural circuits. The results primarily describe the consequences of Foxg1 and Coup-TF1 gain and loss of function experiments on layer IV cell specification, but does not provide any insight into modality-specific circuit development. In fact, it remains unclear if experiments were consistently performed in somatosensory cortex or also in other cortical areas. Large parts of the discussion are devoted to the timing of cell fate specification and sensory circuit development and disease. The manuscript would benefit from tightening up its focus. Many typos and grammatical errors make this manuscript difficult to read.*

Response: We thank Reviewer 3 for the comments. In this study, we focused on the primary somatosensory area to elucidate the mechanisms of sensory circuit wiring, by focusing on the newly identified Egr-Foxg1-COUP-TFI network. We have modified the manuscript to address the Reviewer's concern (Page 1, Line 4; Page 6, Line 24 to Page 7, Line 1). The manuscript has also been English-edited by the Nature Research Editing Service.

6) *Changes in the projection patterns of Coup-TF1 knock-out neurons (Figure 3i-k) should be quantified. What is the effect of Foxg1 overexpression on layer IV cell projection targets?*

Response: As suggested by both of Reviewer 1 and 3, we have now quantified the callosal axons based on the quantification method by Poo and colleagues (Zhou et al., 2013). As shown in the procedures in the new Figure S3, we quantified the signals for callosal projections by selecting a paired region in each condition and analyzed the signal intensity using ImageJ software (Zhou et al., 2013). Cell number was counted using Particle Analysis function and axon signal intensity was then normalized to cell number and shown as bar chart in revised Figure 3l (please refer to the revised Materials and methods 'Quantitative analysis of callosal axons' section for detailed image processing procedure in Page 10, Line 4-18). Based on this analysis, we identify increased callosal projections in the *COUP-TFI* KO cortex, which supports that COUP-TFI removal attenuates layer 4 fate and promotes callosal projection neuronal identity (revised Figure 3i-l). We further performed the same quantitative analysis to assess the impact of Foxg1 acquisition on callosal projections. The results showed increase in callosal projections in the Foxg1 GOF cortex (revised Figure S5d). Together, these results support our finding that Foxg1 and COUP-TFI reciprocally regulates long- and short-range projection identity.

Minor points:

7) *It would be useful if the authors could provide a summary scheme of the genetic interactions that control layer IV cell fate specification.*

Response: We thank the Reviewer for the suggestion. We have now included a summary schematic diagram concerning the genetic interactions that mediate cortical projection subtype specification in the new Figure 10.

8) *Histological images should be presented and labeled in a consistent manner, to allow for comparison and better orientation. In Figure 2, for example, 4 different magnifications are used, none of them are placed into a broader context to facilitate orientation.*

Response: We have modified Figures 2b and 2k to include both low and high magnification views for comparison purposes.

9) *Figure 8 provides evidence for a role of TGF beta / Egr1 signaling in controlling Foxg1 expression in layer IV precursor cells. The experiments that support this conclusion are less comprehensive than the experiments to probe the functional interactions between Foxg1 and Coup-TF1, and the results open up several new questions that remain unanswered. I suggest to focus on Foxg1/Coup-TF1 and address TGF beta / Egr1 signaling in more detail elsewhere. FACS experiments are not described in the Methods.*

Response: We thank Reviewer 3 for raising this critical point. As it was also recommended by other reviewers, we have now performed experiments to examine the consequence of Egr loss-of-function (LOF) (new Figure 9a-l), as well as functional analysis of the predicted Egr-targeted region on the *Foxg1* gene using the CRISPR/Cas9 system (new Figure 9m-v).

To test whether Egr regulates layer 4 cell fate, we disrupted either of *Egr1* or *Egr2* gene and labeled these future layer 4 cells with GFP by E13.75 *in utero* electroporation. The results analyzed at P7 (new Figure 9a-l) showed that in line with other experiments, the majority of GFP-positive cells in the control cortex were located in layer 4 region exhibiting spiny stellate morphology and expressed Ror β (Figure 9b, c, c', f, f', i, i', l). In turn, *Egr1* and *Egr2* deficient cells were shifted in their positions (Figure 9b), showed pyramidal morphology, lost Ror β expression and gained Brn2 expression (Figure 9c-l). Notably, *Egr1*-deficient cells were shifted towards the deeper region, as compared to *Egr2*-deficient cells which were excluded from the barrels (Figure 9b, interaction $p < 0.0001$ by two-way ANOVA), similar to that observed in COUP-TFI-deficient (Figure 3) and *Foxg1*-overexpressed cells (Figure 4e). In addition to this observation, some of *Egr1*-deficient cells gained Ctip2 expression, which was not observed in *Egr2*-deficient cells (Figure 9l). These results indicate differential roles of *Egr1* and *Egr2* on layer 4 differentiation.

As Egr is predicted to target the *Foxg1* promoter to suppress its expression (Figure 8f), we further tested whether presumptive Egr-targeted *Foxg1* promoter region is responsible for Egr mediated *Foxg1* regulation in cell fate selection (Figures 8e, 9m). To assess this, we disrupted the Egr-targeted *Foxg1* promoter region using the CRISPR/Cas9 system in future layer 4 cells, and analyzed the results two days later (new Figure 9n) and at P7 (new Figure 9o-v). Two days after introduction of gRNAs, upregulation of *Foxg1* was observed in these cells indicating that Egr targeted region is responsible for controlling *Foxg1* expression (new Figure 9n). These cells were excluded from layer 4 and showed pyramidal morphology (new Figure 9o) and lost Ror β expression (new Figure 9q, q', v) and gained *Satb2* expression (new Figure 9s, s', v) compared to control cells that were positioned in somatosensory barrels with spiny stellate morphology and expressed Ror β (new Figure 9o, o', p, p'). The effect of disrupting Egr-targeted *Foxg1* promoter region on attenuating layer 4 fate was similar to that of disrupting *Egr2* expression (new Figure 9a-l), *Foxg1*-overexpression (Figure 4a-o) and COUP-TFI LOF studies (Figure 3). In turn, *Egr2*-overexpression (Figure 8i-x), conditional *Foxg1* disruption (Figure 7) and COUP-TFI-overexpression (Figure 6) resulted in layer 4 fate acquisition. Collectively, the results identify a novel Egr-*Foxg1*-COUP-TFI regulatory network in early selection of short- versus long-range projection neuron fate.

The spatial *Egr2* expression pattern in the embryonic cortex further helped us to identify the critical time window required for projection neuron subtype fate selection. Therefore, we performed additional *Foxg1* GOF experiments using the *NeuroD1* promoter, and found that *NeuroD1* driven *Foxg1* overexpression hampers layer 4 fate acquisition, indicating that *Foxg1* in general inhibits layer 4 fate acquisition (revised Figure 4a-o). However, conditional COUP-TFI overexpression using the same *NeuroD1* promoter did not convert layer 5a cells to acquire layer 4 fate (new Figure S9), suggesting the requirement of COUP-TFI prior to *NeuroD1* expression. Considering the positions of GFP cells at the timing of doxycycline treatment in the conditional *Foxg1* knockdown experiments (revised Figure S8c, d), and the spatiotemporal expression of *Egr2* (Figure 8h, h'), *Foxg1* and COUP-TFI (Figure 1e-e''), together with the reported *NeuroD1* expression (Hevner et al., 2006), *Neurog2* expression (Britz et al., 2006) and transcriptomic profile during neurogenesis (Telley et al., 2016) the timing of high-*Egr2* and low-*Foxg1* expression coincides with *Tbr2* expression prior to *NeuroD1* peak expression (new Figure S11). This *Egr2*-*Foxg1* regulation causes subsequent COUP-TFI upregulation, which may explain why COUP-TFI GOF by the *NeuroD1* promoter is too late to promote layer 4 fate. Taken together, the results indicate that the preselection of short- and long-range projection neuron fate commences at early postmitotic stage. At this stage, *Egr2* suppresses *Foxg1* expression, which results in COUP-TFI upregulation to specify layer 4 fate. We hope these lines of evidence is sufficient to position Egr upstream of this regulatory network, and thus have included these new data in Figures 4a-o, 9, S9 and S11, and in the revised manuscript (Page 17 Line 3-6; Page 23, Line 21 to Page 25, Line3; Page 28, Line 11-25).

With regards to the FACS experiments, we had included the cell isolation procedure in the revised Materials and methods section (Page 11, Line 8-13).

References:

- Britz, O., Mattar, P., Nguyen, L., Langevin, L.M., Zimmer, C., Alam, S., Guillemot, F., and Schuurmans, C. (2006). A role for proneural genes in the maturation of cortical progenitor cells. *Cereb Cortex 16 Suppl 1*, i138-151.
- Faedo, A., Tomassy, G.S., Ruan, Y., Teichmann, H., Krauss, S., Pleasure, S.J., Tsai, S.Y., Tsai, M.J., Studer, M., and Rubenstein, J.L. (2008). COUP-TFI coordinates cortical patterning, neurogenesis, and laminar fate and modulates MAPK/ERK, AKT, and beta-catenin signaling. *Cereb Cortex 18*, 2117-2131.
- Hanashima, C., Li, S.C., Shen, L., Lai, E., and Fishell, G. (2004). Foxg1 suppresses early cortical cell fate. *Science 303*, 56-59.
- Hanashima, C., Shen, L., Li, S.C., and Lai, E. (2002). Brain factor-1 controls the proliferation and differentiation of neocortical progenitor cells through independent mechanisms. *J Neurosci 22*, 6526-6536.
- Hatanaka, Y., Namikawa, T., Yamauchi, K., and Kawaguchi, Y. (2016). Cortical Divergent Projections in Mice Originate from Two Sequentially Generated, Distinct Populations of Excitatory Cortical Neurons with Different Initial Axonal Outgrowth Characteristics. *Cereb Cortex 26*, 2257-2270.
- Hevner, R.F., Hodge, R.D., Daza, R.A., and Englund, C. (2006). Transcription factors in glutamatergic neurogenesis: conserved programs in neocortex, cerebellum, and adult hippocampus. *Neuroscience research 55*, 223-233.
- Jabaudon, D., Shnyder, S.J., Tischfield, D.J., Galazo, M.J., and Macklis, J.D. (2012). RORbeta induces barrel-like neuronal clusters in the developing neocortex. *Cereb Cortex 22*, 996-1006.
- Klingler, E., De la Rossa, A., Fievre, S., Devaraju, K., Abe, P., and Jabaudon, D. (2019). A Translaminar Genetic Logic for the Circuit Identity of Intracortically Projecting Neurons. *Curr Biol*.
- Kumamoto, T., Toma, K., Gunadi, McKenna, W.L., Kasukawa, T., Katzman, S., Chen, B., and Hanashima, C. (2013). Foxg1 coordinates the switch from nonradially to radially migrating glutamatergic subtypes in the neocortex through spatiotemporal repression. *Cell reports 3*, 931-945.
- Miyoshi, G., and Fishell, G. (2012). Dynamic FoxG1 expression coordinates the integration of multipolar pyramidal neuron precursors into the cortical plate. *Neuron 74*, 1045-1058.
- Telley, L., Govindan, S., Prados, J., Stevant, I., Nef, S., Dermitzakis, E., Dayer, A., and Jabaudon, D. (2016). Sequential transcriptional waves direct the differentiation of newborn neurons in the mouse neocortex. *Science 351*, 1443-1446.
- Toma, K., Kumamoto, T., and Hanashima, C. (2014). The timing of upper-layer neurogenesis is conferred by sequential derepression and negative feedback from deep-layer neurons. *J Neurosci 34*, 13259-13276.
- Zhou, C., Tsai, S.Y., and Tsai, M.J. (2001). COUP-TFI: an intrinsic factor for early regionalization of the neocortex. *Genes Dev 15*, 2054-2059.
- Zhou, J., Wen, Y., She, L., Sui, Y.N., Liu, L., Richards, L.J., and Poo, M.M. (2013). Axon position within the corpus callosum determines contralateral cortical projection. *Proc Natl Acad Sci U S A 110*, E2714-2723.

Reviewers' Comments:

Reviewer #1:

Remarks to the Author:

The authors performed all the experiments I suggested and corrected all minor points. I think that the quality of the manuscript has improved significantly.

I think it can be published now.

Reviewer #2:

Remarks to the Author:

In this revised manuscript, the authors have made a good effort on addressing my previous concerns. The manuscript is improved. I support the publication.

I noticed that the authors did not cite a few critical references on COUP-TF1 (Armentano et al., Nat Neurosci 2007; Tomassy et al., PNAS 2010; Alfano et al., Development 2011). The authors should cite these references and discuss any differences in result and interpretation.

Reviewer #3:

Remarks to the Author:

This is an extensively and carefully revised manuscript, the authors have addressed all my major concerns. The results presented in this manuscript provide important new insight into the genetic control of cortex development. I recommend publication in Nature Communication.

Point-By-Point Response To The Referees' Comments:

Reviewer #1 (Remarks to the Author):

The authors performed all the experiments I suggested and corrected all minor points. I think that the quality of the manuscript has improved significantly. I think it can be published now.

Response: We thank Reviewer 1 for favorable comments.

Reviewer #2 (Remarks to the Author):

In this revised manuscript, the authors have made a good effort on addressing my previous concerns. The manuscript is improved. I support the publication.

I noticed that the authors did not cite a few critical references on COUP-TF1 (Armentano et al., Nat Neurosci 2007; Tomassy et al., PNAS 2010; Alfano et al., Development 2011). The authors should cite these references and discuss any differences in result and interpretation.

Response: We thank Reviewer 2 for favorable comments and suggestions. We have added these references in the revised manuscript in Page 9, Line5.

Reviewer #3 (Remarks to the Author):

This is an extensively and carefully revised manuscript, the authors have addressed all my major concerns. The results presented in this manuscript provide important new insight into the genetic control of cortex development. I recommend publication in Nature Communication.

Response: We thank Reviewer 3 for favorable comments.